# A Link between the Ice Nucleation Activity and the Biogeochemistry of Seawater

Martin J. Wolf[1,2], Megan Goodell[1], Eric Dong[3], Lilian A. Dove[1,4], Cuiqi Zhang[1,5], Lesly J. Franco[1], Chuanyang Shen[1,6], Emma G. Rutkowski[1], Domenic N. Narducci[7], Susan Mullen[1,8], Andrew R. Babbin[1], and Daniel J. Cziczo[1,9,10]

[1]Department of Earth, Atmospheric, and Planetary Sciences, Massachusetts Institute of Technology, 77 Massachusetts Avenue, Room 54-918, Cambridge, Massachusetts 02139
[2]Yale Center for Environmental Law and Policy, Yale School of the Environment, Yale University, G32 Kroon Hall, 195 Prospect Street, New Haven, CT 06511.
[3]Brown University, 75 Waterman St, Providence, RI 02912
[4]Division of Geological and Planetary Sciences, California Institute of Technology, 1200 East California Boulevard, Pasadena, California 91125
[5]School of Energy and Power Engineering, Beihang University, Beijing, China
[6]Department of Atmospheric and Oceanic Sciences, Peking University, Beijing, China
[7]Department of Biological Engineering, Massachusetts Institute of Technology, 77 Massachusetts Avenue, Room 56-651, Cambridge, Massachusetts 02139
[8]Department of Earth and Planetary Sciences, University of California Berkeley, 307 McCone Hall, Berkeley, California 94720
[9]Department of Chemical Engineering, Massachusetts Institute of Technology, 77 Massachusetts Avenue, Room 66-350, Cambridge, Massachusetts 02139
[10]Department of Earth, Atmospheric, and Planetary Sciences, Purdue University, 550 Stadium Mall Drive, West Lafayette, Indiana 47907

*Correspondence to*: Daniel J. Cziczo (djcziczo@purdue.edu)

**Abstract.** Emissions of ice nucleating particles (INPs) from sea spray can impact climate and precipitation by changing cloud formation, precipitation, and albedo. However, the relationship between seawater biogeochemistry and the ice nucleation activity of sea spray aerosols remains unclarified. Here, we demonstrate a link between the biological productivity in seawater and the ice nucleation activity of sea spray aerosol under conditions relevant to cirrus and mixed-phase cloud formation. We show for the first time that aerosol particles generated from both subsurface and microlayer seawater from the highly productive Eastern Tropical North Pacific Ocean are effective INPs in the deposition and immersion freezing modes. Sea water particles of composition similar to subsurface waters of highly productive regions may therefore be an unrealized source of effective INPs. In contrast, the subsurface water from the less productive Florida Straits produced less effective immersion mode INPs and ineffective depositional mode INPs. These results indicate that the regional biogeochemistry of seawater can strongly affect the ice nucleation activity of sea spray aerosol.

# 1 Introduction

Atmospheric ice nucleation strongly affects the Earth's climate (Pruppacher and Klett, 1980). Cloud albedo and lifetime are altered by ice nucleation processes, impacting the global radiative budget. For instance, ice nucleation changes the size and concentration of cloud particles (Kanji et al., 2017; Pruppacher and Klett, 1980). Clouds comprised of more and smaller ice crystals have comparatively higher albedo than those with larger and fewer ice crystals (Twomey, 1977). The net radiative effect of ice nucleation in clouds depends on several factors, such as convection velocities and the resulting ice crystal concentration (Zhao et al., 2019). Ice formation in mixed-phase clouds is important for initiating precipitation. Ice crystals grow by scavenging water vapor from liquid droplets through the Wegener-Bergeron-Findeisen process, increasing the settling velocities of ice particles (Pruppacher and Klett, 1980). This effect decreases cloud lifetime and is responsible for over 70% of precipitation globally (Lau and Wu, 2003). In these ways, ice nucleation exerts an important impact on the Earth's climate.

Ice nucleation occurs through two main processes. Homogeneous freezing occurs when ice forms spontaneously from any aqueous aerosol. This process requires temperatures below –36 ˚C and relative humidities of at least ~140% with respect to ice (Koop et al., 2000). In the presence of ice nucleating particles (INPs), ice can also form at a lower relative humidity and warmer temperatures through heterogeneous ice nucleation mechanisms (Andronache, 2018; Pruppacher and Klett, 1980). Several pathways of heterogeneous ice formation exist. Depositional ice nucleation occurs above ice saturation but below liquid water saturation when ice deposits directly onto the solid surface of an INP. Depositional ice nucleation and homogeneous freezing are the two predominant pathways for cirrus cloud formation (Barahona et al., 2010; Cziczo et al., 2013; Kärcher, 2017; Lohmann et al., 2004). Immersion freezing can occur above liquid water saturation when an INP first activates as a cloud condensation nucleus. This process is important for ice formation in mixed-phase clouds (Murray et al., 2012; Pruppacher and Klett, 1980). Soluble materials are generally ineffective INPs above liquid water saturation (Hoose and Möhler, 2012). However, organic macromolecules have been demonstrated to promote heterogeneous ice nucleation in solution (Pummer et al., 2015). Such substances are known as ice nucleating macromolecules (INMs) (Vali et al., 2015).

Despite their climatic importance, the sources and characteristics of atmospherically relevant INPs remain uncertain. Laboratory and field studies have identified several terrestrially-sourced INPs (Hoose and Möhler, 2012; Kanji et al., 2017). Characterizing marine sources of INPs also remains an active area of research (Brooks and Thornton, 2018; Kanji et al., 2017). Early studies quantified the ability of sea spray aerosol (SSA) in the marine boundary layer to activate as immersion freezing INPs. Bigg (1990) observed regional differences in ambient INP concentration over the Southern Ocean, but was unable to discern whether the variability resulted from terrestrial influence or from differences in local biological productivity. Rosinski et al. (1986, 1987) performed measurements in the Eastern Tropical Pacific Ocean, finding that local variability in ambient INP concentration corresponded to intensity of upwelling. These studies thereby identified a tentative causality between marine productivity and INP emission rates, but the link between productivity and INP activity remains understudied.

More recently, Wang et al., 2015 and DeMott et al., 2016 simulated blooms in a laboratory setting by co-culturing phytoplankton and heterotrophic bacteria. A positive correlation between primary productivity, dissolved organic matter concentration, and INP emission in these experiments suggested changes in seawater chemistry induced by metabolic activity and grazing can impact the ice nucleation activity of SSA. Wilson et al. (2015) observed that the ice nucleation potential of SSA was correlated to the organic content of aerosols generated from North Atlantic seawater samples. Film burst SSA generated from the organically enriched sea surface microlayer decreased the critical ice supersaturation – the supersaturation at which nucleation initiates – by 10 to 28% compared to more inorganic particles generated from subsurface water (Wilson et al., 2015). Based on the geography and timing of the sample collection, the authors proposed diatom exudates were responsible for the observed deposition and immersion mode nucleation (Knopf et al., 2011). A variety of other phytoplankton, including *Prochlorococcus*, *Synechococcus*, and pico- and nano-eukaryotes, are also effective sources of deposition and immersion mode INPs (Ladino et al., 2016; Wilbourn et al., 2020; Wolf et al., 2019).

Several studies have sought to clarify the importance of marine versus terrestrial INP sources. Ladino et al., 2019 reported that biological particles of possible marine origin were an important source of warm-temperature immersion INPs at a tropical site on the Gulf of Mexico. In contrast, Gong et al., 2020 investigated sources and concentrations of INPs in the seawater and atmosphere near the Cape Verde Islands, finding that SSA was only a minor source of INPs in this region. McCluskey et al., 2018b found that enhanced primary productivity does not necessarily enhance the concentration of INPs in the marine boundary layer. Other studies have sought to parameterize and model the ice nucleation activity of marine INPs. A recent parameterization by McCluskey et al., 2018a demonstrates that nascent SSA exhibits 1/1000[th] of the ice nucleating active sites per unit surface area compared to mineral dust. Global model outputs indicate that SSA may nonetheless be an important source of INPs in remote regions away from terrestrial aerosol inputs (Burrows et al., 2013; Vergara-Temprado et al., 2016).

The formation of SSA encompasses a range of physical processes that affect ice nucleation ability. Winds in excess of 4 m s$^{-1}$ induce whitecaps, which entrain bubbles below the ocean surface (O'Dowd and de Leeuw, 2007). These bubbles accumulate hydrophobic or amphiphilic organic matter as they rise towards the surface (Wilson et al., 2015). Bubble bursting at the surface ejects smaller and organically-enriched film burst particles (Wang et al. 2017; Wolf et al. 2019). The depression in the ocean's surface left by the burst bubble then rapidly fills with the subsurface seawater, ejecting larger jet droplets (Pruppacher and Klett, 1980). The film burst and jet drop mechanisms can produce aerosols with distinctive chemical characteristics. This disparate composition results from differences in the surface activity – that is, the propensity of a molecule to go to the air-sea interface – of organic molecules. SSA particles produced from jet drops are composed mainly of inorganic salts but may also contain whole or fragments of cells and organic molecules with a low propensity to accumulate at the air-sea interface (Wilson et al., 2015; Wolf et al., 2019). Film burst particles can contain higher mass fractions of high surface-activity organic molecules from the sea surface microlayer (Cochran et al., 2017). These natural bubble bursting mechanisms contrast with laboratory methods of aerosolizing seawater. For instance, atomization – a technique employed in this study as well as previous ice nucleation studies (Ladino et al., 2016; Wilson et al., 2015) – does not result in the aerosolization of a microlayer and can result

in aerosol particles with different compositions and size distributions to ambient SSA. The atomization technique employed in this study is further described in the methodology section below.

Research indicates a complex relationship between seawater biogeochemistry and the composition of SSA. Several recent field studies have indicated that rates of primary biological productivity have only a minor influence on the organic content of sea spray (Bates et al., 2020; Quinn et al., 2014; Russell et al., 2010). Other studies have indicated that aged organic matter, such as that metabolized by heterotrophic bacteria, are effectively transferred to the aerosol phase (Cochran et al., 2017; Wang et al., 2015). However, Beaupré et al., 2019 determined that up to 40% of the organic carbon in sea spray could be highly aged, and that the composition of SSA could be less strongly influenced by rates of primary biological productivity in the underlying seawater. Other studies have found that the organic enrichment of SSA is attributable to freshly produced fixed carbon, and that SSA carbon content is correlated with chlorophyl concentration (Ceburnis et al., 2016; O'Dowd et al., 2015). Aside from organic mass fraction, seawater biogeochemistry can also affect the speciation of organic molecules in SSA. Regions of high primary productivity, such as upwelling environments or springtime phytoplankton blooms, exhibit different planktonic species than regions with low primary productivity (Righetti et al., 2019). Whereas upwelling zones and highly productive regions support larger phytoplankton species like diatoms and dinoflagellates, oligotrophic waters are characterized by different clades such as *Prochlorococcus* and *Synechococcus* (Chisholm et al., 1988; Dutkiewicz et al., 2020). Which organisms dominate within the water directly impacts the types of organic molecules and vesicles exuded into the seawater (Azam and Malfatti, 2007; Bertilsson et al., 2005; Biller et al., 2014). Marine regions of high primary productivity are generally enriched in INPs (Wilbourn et al., 2020). INPs from organically-enriched marine waters require lower relative humidities and warmer temperatures to initiate ice nucleation (McCluskey et al., 2017; Wilson et al., 2015).

The impact of ocean biogeochemistry on the ice nucleation activity of SSA remains an active area of research. Studies must investigate the cloud nucleation potential of SSA from diverse marine environments, including coastal, remote, high latitude, tropical, oligotrophic, and eutrophic ecosystems (Brooks and Thornton, 2018; Burrows et al., 2013). DeMott et al., 2016 investigated the ice nucleation activity of seawater from several remote locations, including the Caribbean, the oligotrophic Pacific, and the Bering Sea. Several other studies have focused on high latitude oceans, including the North Atlantic (Wilbourn et al., 2020; Wilson et al., 2015), Arctic (Ickes et al., 2020; Irish et al., 2017), and Southern Oceans (McCluskey et al., 2018b). Gong et al., 2020 found that INPs were both enriched and depleted in the sea surface microlayer relative to subsurface water near the Cape Verde Islands, indicating the effects of both transient biological activity as well as physical parameters such as ocean mixing. Creamean et al., 2019 demonstrated that biological productivity can influence INP concentrations in remote locations when organic material is transported along oceanic currents. These findings indicate the need to understand the sources and abundances of INPs in a diversity of marine environments.

In this study, we identify a link between primary productivity in marine environments and the INP activity of particles generated from seawater in a laboratory setting, which we refer to as sea water particles (SWPs). Two chosen sample regions – the Florida Straits and the Eastern Tropical North Pacific (ETNP) – are typical of low productivity and

highly productive marine ecosystems, respectively. Coastal upwelling along the eastern boundary of the Pacific sustains high levels of primary productivity in the ETNP. We demonstrate that both the subsurface and microlayer seawater can be sources of effective INPs in highly productive marine environments. Our findings show for the first time that aerosols formed from subsurface waters in productive regions can be effective INPs. These results demonstrate that SWP composition and INP activity varies between marine biogeochemical environments, yielding important caveats for climate models parameterizing marine INP impacts on global climate.

## 2 Experimental Methods

### 2.1 Sampling Locations

Seawater samples and seawater measurements were taken on two cruises to the Florida Straits and ETNP. Sampling in the Florida Straits took place aboard the SSV *Corwith Cramer* from March 28[th] through March 31[st] 2018. Sampling in the ETNP took place aboard the R/V *Falkor* from June 30[th] through July 10[th] 2019 (Figure 1, Table S1). At each location, microlayer and subsurface samples were collected. Additional context for these samples was gained through analysis of marine biogeochemical parameters like nitrate, phosphate, pH, and chlorophyll (Table 1). These variables were measured using standard methods (Braman and Hendrix, 1989; Clayton and Byrne, 1993; Evans et al., 2020; Strickland and Parsons, 1972).

### 2.2 Seawater Sampling

The sea surface microlayer was sampled using the glass plate technique detailed previously (Harvey and Burzell, 1972; Irish et al., 2017). Briefly, a plexiglass plate was fully submerged under seawater and withdrawn at a rate of approximately 5 cm s$^{-1}$, allowing microlayer organics to adhere to the plate (Figure S1). The withdrawn plate was allowed to drain for 5 seconds, and then was scraped dry using a neoprene wiper blade cleaned with isopropanol between samples. The sampled microlayer was collected in acid washed 250 mL Nalgene bottles rinsed with subsurface seawater from the sampling station. Approximately 200 mL was collected for each seawater sample, requiring an average of 102 dips per sample. Sampling occurred on the windward side of the ship to avoid contamination. Although previous studies have sometimes sampled as far as 500 m away from the ship (Irish et al., 2017; Wilson et al., 2015), rough seas precluded this practice on our cruises. Wind speed averaged 13.5 and 15 m s$^{-1}$ in the Florida Straits and ETNP, respectively, and at times exceeded 20 m s$^{-1}$ (Table 1). The resulting rough seas could possibly have impacted sea surface microlayer characteristics. For instance, Rahlff et al., 2017 determined that bacterial enrichment in the sea surface microlayer occurred only at winds speeds below approximately 5 m s$^{-1}$. Further studies have also determined a link between wind speed and the composition of the sea surface microlayer. Sun et al., 2018 determined that the abundance and size of macromolecular gels in a wave tank's microlayer decreased with winds above 8 m s$^{-1}$. Other studies have found an organic enrichment in the microlayer with wind speeds ranging from 10 to 13 m s$^{-1}$ (Sabbaghzadeh et al., 2017; Wurl et al., 2011), indicating that conditions were at times conducive to microlayer formation during our sampling.

Subsurface seawater samples were collected at the same time and location as the microlayer samples with a Seabird conductivity-temperature-depth  rosette. Seawater was sampled from the shallowest Niskin bottle on each cast and typically between 2 and 5 meters below the surface. Subsurface samples were collected in sterilized 250 mL Nalgene bottles rinsed with seawater from the same Niskin that was sampled. Both subsurface and microlayer waters were stored at -80 ˚C until analysis. Previous analysis suggests that freezing seawater samples has minimal effect on INPs (Schnell and Vali, 1975).

**2.3 Seawater Aerosolization**

To investigate the chemical and ice nucleating properties of aerosols generated from collected seawater, samples were thawed by immersing sealed bottles in room temperature water and mixed by inverting bottles ten times. 50 mL of sample were added to a glass container attached to a custom collision-type atomizer. The atomizer is constructed from machined aluminum and is based on the design of the TSI Model 3076 constant output atomizer (TSI, 2005). Briefly, filtered pressurized (30 psi) air is passed through a 0.01 inch critical orifice. Following the orifice, the air expands, causing seawater sample to be drawn up through inert polyethylene tubing and atomized by the jet of air. A polydisperse aerosol particle stream with a constant number and size distribution is created by the atomizer (Figure S2). The atomizer and tubing were sonicated with deionized water between samples to avoid cross-contamination.

We note that the atomization technique – although used in prior studies investigating the ice nucleation of sea spray aerosol (Ladino et al., 2016; Wilson et al., 2015) – has several limitations. Specifically, atomization produces aerosol with different physical and chemical characteristics than ambient SSA (Collins et al., 2014). First, atomization results in a different aerosol size distribution due to the lack of bubble bursting mechanisms (Fuentes et al., 2010). The impact of this artefact can be limited by size-selecting a narrow diameter range from the resulting polydisperse aerosol stream prior to INP analyses. However, atomization also produces aerosols of a different composition than ambient SSA (Gaston et al., 2011). Natural bubble-bursting mechanisms result in aerosol with size-dependent composition (Collins et al., 2014; O'Dowd et al., 2004; Prather et al., 2013). It is unlikely that atomization can replicate the composition of natural SSA. Further, atomization is an energetic process that may result in a higher rate of cell lysis than expected from natural processes, such as apoptosis, viral infection, or predator grazing (Agustí and Duarte, 2013; Kirchman, 1999). This may artificially increase the organic content of our laboratory-generated aerosol and increase the occurrence of ice nucleating macromolecules in particles (Ickes et al., 2020; Knopf et al., 2011). We also note that the atomizer draws seawater from below the surface. Although our thawed seawater samples were homogenized with vigorous shaking prior to atomization, organic partitioning at the surface occurs rapidly. Cunliffe et al., 2013 observed that the composition and bacterial makeup of microlayer samples were reestablished only minutes after disruption. We therefore acknowledge the limits of our laboratory-generated data when it comes to drawing conclusions about ambient processes.

Aerosols were dried by passing through two consecutive diffusion dryers filled with silica desiccant. Relative humidity at the outlet of the diffusion dryers was 15%, which is below the efflorescence relative humidity of sea salt (Cziczo

and Abbatt, 2000; Zeng et al., 2013).  The resulting dried sea salt aerosols were diverted into a differential mobility analyzer (DMA, Model 2002; Brechtel Manufacturing Inc., Hayward, CA). Particles were size selected (mobility diameter = 200 nm) with a sheath to sample flow ratio of 8:1 (Figure S2). The DMA sheath flow was dried with silica desiccant to a relative humidity of less than 15%. A 500 nm size cutoff impactor was used upstream of the DMA to

prevent large multiply-charged particles from entering the sampled aerosol stream. Nonetheless, doubly charged particles may have been sampled. Figure S2 illustrates that the concentrations of doubly and triply charged particles (approximately 330 and 450 nm in diameter, respectively) are much lower than singly charged particles. Concentrations of particles with multiple charges were less than 20% of the concentrations of 200 nm particles we size selected. Given that the ratio of doubly to singly charged particles predicted by a Fuchs charging model applied

to a DMA neutralizer is below 0.3 (Mamakos, 2016), we estimate that multiply charged particles constitute only less than 6% of the total particles sampled. The 200 nm particle diameter was chosen to align with previous experiments' methods (DeMott et al., 2016; Wilson et al., 2015), yet we acknowledge INP activity varies with SSA diameter (DeMott et al., 2016; Si et al., 2018). This choice reflects previous findings that marine INPs are likely macromolecular organic clusters smaller than 200 nm. For instance, Irish et al., 2017 quantified INP size in Arctic seawater samples,

identifying that the majority of immersion-mode INPs in seawater were between 20 and 200 nm. Wolf et al., 2019 further demonstrated that a variety of marine-derived molecules smaller than 200 nm were INP active in the depositional ice nucleation mode. A likely source of these molecular INPs are phytoplankton exudates (Ickes et al., 2020; Knopf et al., 2011; Wilson et al., 2015).

**2.4 Chemical Characterization**

We investigated the composition of 200 nm SWPs generated from seawater samples using the Particle Analysis by Laser Mass Spectrometry (PALMS) instrument (Cziczo et al., 2006). PALMS measures mass spectra on a particle-by-particle basis, allowing the composition of aerosols from like sources to be compared. Sampled particles are first collimated in an aerodynamic inlet. The carrier gas is pumped away under vacuum, yet the residence time before

ionization is short enough to minimize the loss of volatile organic components from the particulate surface (Cziczo et al., 2006).

Particles are then ionized using a 193 nm ultraviolet excimer laser. Atomic and small molecular ions are then sampled using time of flight mass spectrometry (Murphy, 2007). PALMS measures either positive or negative mass spectra

30    per particle. Although an organic signal is detected in both polarities, sampling in the negative polarity captures more organic nitrogen and phosphate markers (Wolf et al., 2019). We sampled approximately 2,000 particles in the negative polarity for each seawater sample.  Particle ionization with the UV excimer is not quantitative (Cziczo et al., 2006; Murphy et al., 1998). However, the average relative intensity of organic signal in a sample's mass spectra can qualitatively indicate which seawater samples are organically-enriched (Wolf et al., 2019).

**2.5 Ice Nucleation Measurement**

The SPectrometer for Ice Nuclei (SPIN; Droplet Measurement Technologies, Boulder, CO) measured the conditions required for the generated SWPs to nucleate ice and the fractional INP activation. The theory and operation of SPIN has been described previously (Garimella et al., 2016, 2017). Briefly, SPIN is a continuous flow diffusion chamber style instrument that simulates ice nucleation conditions in clouds. It consists of two flat parallel plates separated by 1.0 cm and coated in approximately 1 mm of ice.

Size-selected aerosol particles were drawn into the nucleation chamber and nominally constrained to a flow centerline with particle free sheath air adjacent to the ice-covered walls. The temperature and relative humidity that the aerosols experience is controlled by varying the temperature gradient between the two walls (Garimella et al., 2016; Kulkarni and Kok, 2012). In this study, SPIN operated in two different temperature and ice saturation ratio ($S_{ice}$) regimes. To observe deposition and homogeneous freezing, SPIN's aerosol lamina varied between –40 to –46 ˚C and $1.0 \leq S_{ice} \leq$ 1.6. These conditions are relevant to cirrus cloud formation. To observe immersion freezing, SPIN's aerosol lamina ranged from –20 to –30 ˚C and $1.0\ S_{ice} \leq 1.5$; conditions that can also correspond to liquid water supersaturation and mixed phase cloud formation.

Turbulent mixing near the aerosol inlet causes particles to spread outside of the aerosol lamina. This exposes particles to a wider temperature range and lower $S_{ice}$ than that of the lamina centerline (Garimella et al., 2017). Particles outside of the lamina are therefore less likely to activate as INPs. To account for this artefact, a correction factor is normally applied to measured INP and fractional activation data (DeMott et al., 2015; Garimella et al., 2017; Wolf et al., 2019). We apply the methods and correction factors detailed in Garimella et al. (2017) and Wolf et al. (2019) to immersion and deposition nucleation data presented in this study.

After the nucleation chamber, particles enter an optical particle counter (OPC). The OPC records side scatter intensity and laser light depolarization data on a particle by particle basis for diameters between 0.2 and 15 μm. A machine learning algorithm, detailed in Garimella et al. (2016), is trained using four OPC variables to classify all particles as either unactivated, ice, or liquid droplets. Fractional INP activation is derived by dividing ice crystal concentration assigned by the machine learning output by total particle concentration, as measured by a condensation particle counter (CPC, Model 1700; Brechtel Manufacturing Inc., Hayward, CA) running in parallel to SPIN. Frost shedding from SPIN's iced walls creates a baseline ice crystal concentration. These frost "backgrounds" are measured before and after each experiment. The average value is subtracted from the measured INP concentration. Background concentrations are typically below 10 $L^{-1}$ and below the threshold ice concentration used to determine nucleation onset in all experiments presented herein.

## 3   Results and Discussion

### 3.1 Seawater Chemistry

Analysis of SWPs generated from the two sampled regions suggests that the ocean biogeochemistry impacts the relative composition of subsurface and microlayer waters. We measured the intensity of carbon, organic nitrogen, and

phosphorus signals. The integrated carbon signal from PALMS is defined here as the sum of the areas under the $C^-$ (m/z = 12), $C_2^-$ (m/z = 24), and $C_4^-$ (m/z = 48) mass spectra peaks (Figure 2). The $C_3^-$ (m/z = 36) peak was omitted due to its proximity to two chlorine isotopic peaks (m/z = 35 and 37), the intensity of which vary between spectra. Omitting the $C_3^-$ peak does not affect our analysis, since the ratios of $C_n^-$ peaks are similar across all spectra. Similarly, the integrated nitrogen signal is defined as the sum of $CN^-$ (m/z = 26) and $CNO^-$ (m/z =42) peaks. These peaks may

result from the ionization of amine functional groups, such as those found in amino acids. We omit inorganic nitrogen ions, such as $NO^-$ and $NO_2^-$, as these may result from nitrate salts in SWPs and would not increase the INP activity. An organic phosphorus signal, defined as $CP^-$, was not observed (Figure 2). The phosphorus signal is defined as the sum of $PO_2^-$ (m/z = 63), $PO_3^-$ (m/z = 79), and $PO_4^-$ (m/z = 95). These peaks may indicate the ionization of phospholipids and the phosphate backbones of nucleic acids.

The average integrated carbon, nitrogen, and phosphorus signals (n > 1000 for each data point) are shown in Figure 3, along with ordinary least squares linear regressions. Both the subsurface and microlayer waters of the highly productive ETNP exhibit similar organic carbon signals, indicated by the slope of approximately $1 \pm 0.09$ (standard error; Figure 3a) when comparing the ratio of organic carbon signals in microlayer and subsurface samples. This

suggests that elevated primary productivity in the ETNP sustained organic carbon content in the subsurface waters more so than in the Florida Straits. This aligns with metrics of higher primary productivity in the ETNP subsurface water, such as higher average chlorophyll concentrations at the deep chlorophyll maximum (Table 1). Conversely, a slope greater than 1 for the Florida Straits samples (slope = $3.85 \pm 0.25$ (standard error); Figure 3d) indicates a compositional disparity between SWPs generated from the subsurface and microlayer waters.

The average organic carbon signal for the ETNP subsurface samples was $1.11 \pm 0.62$ ($1\sigma$ variability), whereas the average organic carbon signal for the Florida Straits subsurface samples was $0.41 \pm 0.20$. Reported uncertainty is a standard deviation of variability across spectra signals. Likewise, median values are 0.76 and 0.29 for the ETNP and Florida Straits subsurface samples, respectively. The relatively higher concentration of organics in the ETNP

subsurface water is in agreement with the higher rates of primary productivity there than in the Florida Straits. Organic carbon signal can be a better indicator of primary productivity than concentrations of nutrients like nitrate ($NO_3^-$) and phosphate ($PO_4^{3-}$) and chlorophyll concentration (Table 1). Low nutrient concentrations can indicate that nutrients are being consumed or that they are low to begin with. Further, the chlorophyll to carbon ratio in seawater can vary (Lefèvre et al., 2003). These results agree with previous measurements of seawater composition. For instance, one

study found that microlayer samples were organically enriched in the open ocean but unenriched in coastal upwelling zones similar to the ETNP region sampled here (Zäncker et al., 2017).

Our PALMS analysis did not demonstrate that organic nitrogen or phosphorus preferentially partitioned into the microlayer in either the ETNP or Florida Straits. Both subsurface and microlayer waters in the ETNP and Florida Straits yielded similar organic nitrogen signal intensities (Figures 3b and 3e). Several previous studies have found that amino acids are enriched in microlayer samples relative to subsurface from both coastal and remote waters (Engel and Galgani, 2016; Kuznetsova et al., 2004; Kuznetsova and Lee, 2002; Reinthaler et al., 2008; Zäncker et al., 2017). PALMS detection of more soluble organic nitrogen species in subsurface waters in addition to certain amino acids that partition in the microlayer could have led us to observe parity of organic nitrogen in both microlayer and subsurface samples. Several factors, such as matrix effects and variable ionization efficiencies of different molecules, can affect the observed signal in PALMS mass spectra (Cziczo et al., 2006; Murphy, 2007; Murphy et al., 2006; Zawadowicz et al., 2017). Nitrogenous molecules in subsurface waters can include byproducts of microbial protein degradation, which tend to increase the solubility of nitrogenous molecules (Engel et al., 2018). We also did not observe an enrichment in phosphorus in either the ETNP or Florida Straits microlayer samples (Figures 3c and 3f). Although an enrichment of phosphorus in the microlayer due to lipid partitioning at the air-sea interface is expected, we note that lipids are labile and short-lived in seawater (Kattner et al., 1983; Parrish et al., 1992). Organophosphate groups on lipids are rapidly degraded by bacterial processes, thereby increasing their solubility (Ogunro et al., 2015). This leads phosphorous to be more rapidly recycled compared to carbon and nitrogen nutrients.

Our compositional analysis demonstrates variability in the composition of our laboratory-generated aerosol particles. Aerosols generated from both the subsurface and microlayer samples from the highly productive ETNP contained similar organic contents. The Florida Straits samples indicated a compositional disparity between microlayer and subsurface samples, with SWPs generated from subsurface water depleted in organics.

### 3.2 Deposition Mode Ice Nucleation

To investigate possible links between SWP composition and ice nucleation activity, we quantified the conditions required to initiate ice nucleation in the deposition nucleation regime (T < -40 ˚C). The ice supersaturation at ice nucleation onset in the deposition freezing mode – termed critical supersaturation – is a metric of the activity of ice nucleating substances. INPs that activate at lower supersaturations and warmer temperatures are able to initiate cloud formation over a wider range of atmospheric conditions (Kanji et al., 2017; Pruppacher and Klett, 1980). Keeping with previous studies, we characterize ice nucleation onset as when 1% of particles depositionally nucleate ice (Kanji et al., 2017; Wilson et al., 2015).

Aerosols generated from organically-enriched samples generally required a lower $S_{ice}$ to attain 1% fractional activation than organically depleted samples (Figure 4). Aerosols from both the subsurface and microlayer ETNP samples exhibited a similar critical $S_{ice}$, ranging from about 1.10 to 1.35 (Figure 4a). This finding contrasts with results from the Florida Straits samples, which display divergent INP activity for microlayer and subsurface samples (Figure 4b). Whereas the microlayer samples typically initiated depositional nucleation between $1.13 \leq S_{ice} \leq 1.30$, the subsurface samples often nucleated homogeneously ($S_{ice} > 1.40$). The $S_{ice}$ at onset decreases at a rate of approximately 0.039 and

0.092 per degree cooling for Florida Straits and ETNP microlayer SWPs, respectively. These trends are comparable to previous studies on deposition ice nucleation of organic SSA surrogates (Ladino et al., 2016; Schill and Tolbert, 2014; Wolf et al., 2019).

The range of critical $S_{ice}$ values agrees with results from North Atlantic microlayer samples, as shown in Figure 4b. (Wilson et al. 2015). The critical $S_{ice}$ for microlayer samples at each temperature did not correlate ($R^2 < 0.2$; $p \geq 0.2$) with total carbon, nitrogen, or phosphorus PALMS signal. This suggests individual components of seawater are more important than bulk composition in driving ice nucleation. Candidates may be carbohydrates, individual proteins, and polysaccharides, as these compounds are effective depositional INPs and are enriched in the microlayer (Engel et al.,
2018; Russell et al., 2010; Thornton et al., 2016; Zäncker et al., 2017; Zeppenfeld et al., 2019).

Whereas the North Atlantic subsurface samples in Wilson et al., 2015 did not nucleate heterogeneously, our subsurface samples from the ETNP nucleated at low ice supersaturations ($S_{ice} = 1.1$ at $T = –46$ ˚C). This indicates that SWPs from these biogeochemically distinct regions exhibit different ice nucleation activity. Aerosols generated from subsurface
waters in less productive regions are ineffective depositional INPs. Such aerosols can originate from jet droplets, which are formed when water beneath the microlayer is ejected as bubbles burst (Quinn et al., 2015; Wu, 2002). Conversely, organically-enriched subsurface water from the highly productive ETNP region demonstrate similar critical $S_{ice}$ values as SWPs from microlayer samples (Figure 4a).

Biological productivity in subsurface water can therefore impact the chemical makeup and INP activity of both jet droplet and film burst SSA (Wang et al., 2015). Our measurements of seawater biogeochemistry indicate that the ETNP had characteristics of higher biological activity than the Florida Straits (Table 1). For instance, average chlorophyll concentrations at sampling stations' deep chlorophyll maxima were 0.35 mg/m$^3$ higher in the ETNP than in the Florida Straits sampling stations. Satellite-derived regional surface chlorophyll a concentrations were also nearly
double in the ETNP stations (0.19 mg m$^{-3}$) than the Florida Straits stations (0.10 mg m$^{-3}$) during sampling (Figure S3). This elevated productivity is maintained by higher nutrient concentrations. Nitrate concentrations in subsurface water samples were on average 0.14 µM greater in the ETNP. Nitrogenous nutrients generally limit primary productivity across the tropical and subtropical oceans. However, critical $S_{ice}$ was not directly correlated with metrics of primary productivity or wind speed (Table S2). Even stations with the highest chlorophyll or nutrient concentrations
(Table 1) did not correspond to the lowest critical $S_{ice}$ values. This suggests factors other than primary productivity, such as cell lysis and microbial degradation, likely play an important role in determining the INP activity of SSA (McCluskey et al., 2017). Moreover, the standing stock concentration of nutrients does not necessarily reflect productivity as primary producers can dynamically draw down these concentrations. Other governing factors may include the types of plankton supported by the seawater biogeochemistry. While some species, such as diatoms and
*Prochlorococcus* have been found to be effective sources of depositional INPs, other plankton species are poor sources of INPs (Junge and Swanson, 2008; Knopf et al., 2011; Wolf et al., 2019).

### 3.3 Immersion Mode Ice Nucleation

We observed a similar relationship between seawater biogeochemistry and immersion mode ice nucleation. INP active site densities are defined as the equivalent number of sites that promote ice nucleation per unit particle surface area (Vali et al., 2015). Active site density ($n_s$) is a metric of the effectiveness of different aerosols as INPs (Kanji et al., 2017; Vali et al., 2015). We calculated $n_s$ values for each aerosol sample by dividing the activated immersion mode INP concentration ($n_i$) by the total aerosol surface area concentration ($n_A$):

$$(1) \qquad\qquad\qquad n_s = \frac{n_i}{n_A} = \frac{f_i}{n_a}$$

where $f_i$ is the fractional INP activation in the immersion mode and $n_a$ is the surface area of a single particle. Equation 1 is an approximation applicable to small fractional activations. We use fractional activation values corrected for aerosol spreading outside the central lamina (DeMott et al., 2015; Garimella et al., 2017), as detailed in Section 2.5 above. In deriving $n_a$, it is assumed effloresced SWPs are spherical. Parameterizations of $n_s$ can be size-dependent when aerosol composition varies with size. Atomizing seawater creates aerosol particles less enriched in organics than natural seawater aerosolization processes, as it does not mimic the film burst and jet drop aerosolization processes that create organically-enriched and depleted particles, respectively. For instance, Gaston et al., 2011 observed that atomizing seawater produces over 27% fewer organically-enriched particles compared to bubbling. The majority of 200 nm particles in ambient SSA arise from the film-burst production process (Pruppacher and Klett, 1980). Wang et al., 2017 determined that film-burst particles constitute at least 57% of submicron SSA, with the remainder resulting from jet droplets. Our atomized SWPs are likely less organically-enriched than ambient SSA. However, we note that the atomization process energetically aerosolizes the seawater solution, potentially rupturing cells, resulting in particles with more INP-active organic macromolecules than might occur in natural SSA. At this time, it is unknown whether the atomization technique results in a greater or lesser $n_s$ density compared to natural SSA formation mechanisms.

SWPs generated from both subsurface and microlayer ETNP samples yielded comparable $n_s$ values (Figure 5a). Average $n_s$ values for ETNP microlayer and subsurface samples were indistinguishable within a standard deviation of variability. For instance, the average microlayer $n_s$ at -30 ˚C was $3.3 \pm 1.9 \times 10^5$ cm$^{-3}$, compared with $2.1 \pm 1.3 \times 10^5$ cm$^{-3}$ for subsurface samples. In contrast, the organically-depleted Florida Straits subsurface samples were less effective immersion mode INPs than the organically-enriched microlayer samples (Figure 5b). Values of $n_s$ for these subsurface samples were typically one to two orders of magnitude lower than the average microlayer $n_s$ value at a given temperature. Possible substances causing immersion mode ice nucleation are organic macromolecules. Such INMs include carbohydrates, liposaccharides, and ice nucleating active proteins (Ogunro et al., 2015; Pummer et al., 2015) as well as their byproducts of microbial degradation (McCluskey et al., 2017).

Our $n_s$ values agree well with previous measurements of similarly sized SSA across various marine regions (Figure 5). Values for 250 nm aerosol from productive coastal waters (Si et al., 2018) are closer to the $n_s$ of the organically-

enriched samples from the ETNP and Florida Straits microlayer (Figure 3). Further, organically-depleted samples from the Florida Straits subsurface waters are more in agreement with open ocean measurements (DeMott et al., 2016). This finding demonstrates the importance of seawater biogeochemistry in determining immersion mode INP activity. However, we caution that our derived $n_s$ values should not be used to extrapolate $n_s$ for ambient marine aerosol. SSA may differ in composition to our SWPs, and total marine aerosol includes many particle sources not considered here, such as secondary aerosol sources (Facchini et al., 2008; Fu et al., 2013; O'Dowd and de Leeuw, 2007).

### 3.4 Atmospheric Implications

Our findings demonstrate that SWPs generated from subsurface waters in highly productive marine environments can be comparably effective INPs as aerosols generated from the microlayer. Figure 6 summarizes the ice nucleation activity of SWPs generated from the ETNP and Florida Straits. In the immersion freezing mode, subsurface SWPs from the ETNP demonstrate $n_s$ values orders of magnitudes greater than those from the Florida Straits subsurface (Figure 6a). We also show for the first time that subsurface waters can be an effective source of depositional INPs in highly productive marine environments. Critical $S_{ice}$ values to attain 1% fractional activation for subsurface ETNP samples overlapped with $S_{ice}$ thresholds for microlayer samples (Figure 6b). Samples from less productive regions, such as the Florida Straits and the North Atlantic Ocean (Wilson et al. 2015), did not identify subsurface samples as sources of effective depositional INPs (Figure 6b).

Our results augment previous findings that particle composition determines the ice nucleation activity of SSA (DeMott et al., 2015b; McCluskey et al., 2017; Wilbourn et al., 2020; Wilson et al., 2012; Wolf et al., 2019). We also found that INP activity is uncorrelated with variables like nutrient concentration, wind speed, and chlorophyll concentration (Table S2). This indicates that factors other than rates of primary productivity are also important determinants of SWP composition and INP activity. These processes likely include plankton diversity and microbial degradation of organic components in seawater (DeMott et al., 2016; McCluskey et al., 2017; Wang et al., 2015). These considerations are likely to matter most in environments without significant continental aerosols. For instance, modeling studies indicate that SSA constitute a greater fraction of ambient INP in remote regions free from intrusions of mineral and desert dust aerosol (Burrows et al., 2013; Vergara-Temprado et al., 2016). Several recent field studies have also indicated the potential importance of marine sources of INPs in both remote and coastal atmospheres (Creamean et al., 2019; Ladino et al., 2019; McCluskey et al., 2018a). Highly productive marine regions like the ETNP are generally found near coasts, where terrestrial INP sources most often dominate over marine emissions (Burrows et al., 2013; Sarmiento and Gruber, 2006; Vergara-Temprado et al., 2017). An exception is the Southern Ocean, where austral summer marine primary productivity is high, and ambient atmospheric dust concentrations are low (Jickells, 2005). Despite these factors, recent measurements found low concentrations of immersion-mode INP over the Southern Ocean (McCluskey et al., 2018b). This demonstrates that INP concentrations are not uniformly elevated in highly productive marine environments.

These findings emphasize the heterogeneity of SSA composition, ice nucleation activity, and climatic impact. Smaller film burst particles originating from the sea surface microlayer are generally considered to be the most effective SSA INPs (Wilson et al., 2015; Wolf et al., 2019). Our results further highlight a potential shortcoming of commonly employed model parametrizations that use surface chlorophyll concentrations as a predictor of aerosol organic mass fraction and INP activity (Burrows et al., 2013; O'Dowd et al., 2008; Vergara-Temprado et al., 2016). Despite similar surface chlorophyll a concentrations (Table 1), SWPs generated from the ETNP and Florida Straits yielded different ice nucleation properties. In biologically active marine ecosystems such as the ETNP, primary production in deeper subsurface waters can increase SSA INP concentration and enhance ice nucleation activity.

## 4   Conclusions

Sea spray is the largest aerosol source on Earth by mass, with total global emissions estimated to be $1 - 3 \times 10^{16}$ g yr–1 (Erickson and Duce, 1988; Vignati et al., 2010). Despite these large emissions, the impact of seawater biogeochemistry on SSA composition and INP activity remains uncertain. To clarify the importance of primary productivity in these factors, we measured the composition and ice nucleation activity of SWPs generated from two biogeochemically diverse marine regions. Our seawater samples from the ETNP represent seawater in highly productive marine environments, whereas samples from the Florida Straits are characteristic of less productive environments.

We studied the impact that regional biogeochemistry has on SWP composition and ice nucleation activity. The highly productive ETNP region exhibits similar organic contents in subsurface and microlayer seawater. The Florida Straits microlayer is conversely organically-enriched relative to subsurface water. We then studied the regional differences in SSA ice nucleation activity. SWPs generated from both subsurface and microlayer waters in the ETNP were effective depositional and immersion mode INPs. However, we observed that subsurface SWPs from the Florida Straits were less effective INPs than microlayer SWPs. These results indicate that ocean biogeochemistry plays an important role in the emission of marine INPs. Organically enriched film burst and jet drop aerosol emitted from highly productive marine regions may have a locally greater influence on ice nucleation, precipitation, and radiative budget than emissions from oligotrophic waters.

## 5   Data Availability

Data used to generate this manuscript's figures are included in the Supplemental Information section and a Harvard Dataverse dataset under the name: "A Link between the Ice Nucleation Activity and the Biogeochemistry of Seawater." The DOI of this dataset is: 10.7910/DVN/QEJJMF. Further data inquires can be directed to the corresponding author (Daniel Cziczo, djcziczo@purdue.edu).

## 6   Acknowledgements

The authors declare no competing financial interests. We thank our peer reviewers for their useful comments and suggestions that have improved the manuscript. We further thank the crew, technicians, and scientists onboard the

SSV *Corwith Cramer* and the R/V *Falkor* for their logistical and scientific support during sampling. We also thank the students of the MIT Field Oceanography Course 12.373/12.777 for their assistance sampling onboard the SSV *Corwith Cramer*. Funding was provided in part by an MIT Environmental Solutions Initiative educational grant to ARB. R/V *Falkor* ship-time was awarded by a Schmidt Ocean Institute grant to Karen Casciotti and ARB. The SSV *Corwith Cramer* expedition was funded by the MIT Houghton Fund and by the MIT/WHOI Joint Program.

## 7 Author Contributions

MJW, DJC, and ARB designed the experiments and methodology. MJW, MG, ED, LAD, CZ, LJF, CS, ER, DNN, and SM collected seawater samples, performed chemical analyses, and/or measured ice nucleation activity. MJW, DJC, and ARB prepared manuscript with input from all coauthors.

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

**Table 1 – Seawater Sampling Station Characteristics**

| Region | Station | Wind Speed (m/s) | [NO$_3^-$] (µM) | [PO$_4^{3-}$] (µM) | pH | Surface [Chl.-a] (mg/m$^3$) | Max Deep [Chl.-a] (mg/m$^3$) |
|---|---|---|---|---|---|---|---|
| | **1** | 14.0 | 0.36 | 0.24 | 8.06 | 0.09 | 1.27 |
| | **2** | 13.6 | 0.17 | 0.25 | 8.05 | 0.10 | N/A |
| | **3** | 26.3 | 0.13 | 0.14 | 8.06 | 0.10 | 1.08 |
| | **4** | 18.5 | 0.19 | 0.02 | 8.03 | 0.09 | 1.10 |
| | **5** | 13.1 | 0.18 | 0.19 | 8.06 | 0.08 | 1.19 |
| **Florida Straits** | **6** | 20.3 | 0.60 | 0.10 | 8.05 | 0.07 | 1.18 |
| | **7** | 13.7 | 0.17 | 0.17 | 8.02 | 0.10 | 1.18 |
| | **8** | 12.3 | 0.20 | 0.09 | 8.01 | 0.12 | 1.42 |
| | **9** | 12.5 | 0.15 | 0.06 | 8.04 | 0.09 | 1.49 |
| | **10** | 10.1 | 0.18 | 0.02 | 8.07 | 0.06 | 1.24 |
| | **11** | 11.0 | 0.18 | 0.04 | 8.03 | 0.06 | 0.87 |
| | **Average (± 1σ)** | **15.0 (± 4.8)** | **0.23 (± 0.1)** | **0.12 (± 0.1)** | **8.04 (± 0.02)** | **0.09 (± 0.02)** | **1.20 (± 0.2)** |
| | **1** | 17.8 | 0.00 | 0.30 | 8.08 | 0.08 | 1.22 |
| | **2** | 22.7 | 1.02 | 0.24 | 8.07 | 0.08 | 1.06 |
| | **3** | 17.2 | 0.00 | 0.23 | 8.06 | 0.06 | 1.43 |
| | **4** | 15.3 | 0.00 | 0.22 | 8.07 | 0.06 | 3.34 |
| | **5** | 12.1 | 0.49 | 0.27 | 8.06 | 0.10 | 1.69 |
| **ETNP** | **6** | 11.6 | 0.32 | 0.26 | 8.07 | 0.09 | 1.38 |
| | **7** | 7.9 | 1.34 | 0.39 | 8.03 | 0.09 | 1.38 |
| | **8** | 11.0 | 0.00 | 0.28 | 8.05 | 0.07 | 1.35 |
| | **9** | 10.3 | 0.00 | 0.24 | 8.07 | 0.06 | 0.98 |
| | **10** | 11.6 | 0.00 | 0.25 | 8.07 | 0.06 | 0.67 |
| | **11** | 11.3 | 0.89 | 0.29 | 8.07 | 0.06 | 1.65 |
| | **Average (± 1σ)** | **13.5 (± 4.3)** | **0.37 (± 0.5)** | **0.27 (± 0.05)** | **8.06 (± 0.01)** | **0.07 (± 0.02)** | **1.55 (± 0.7)** |

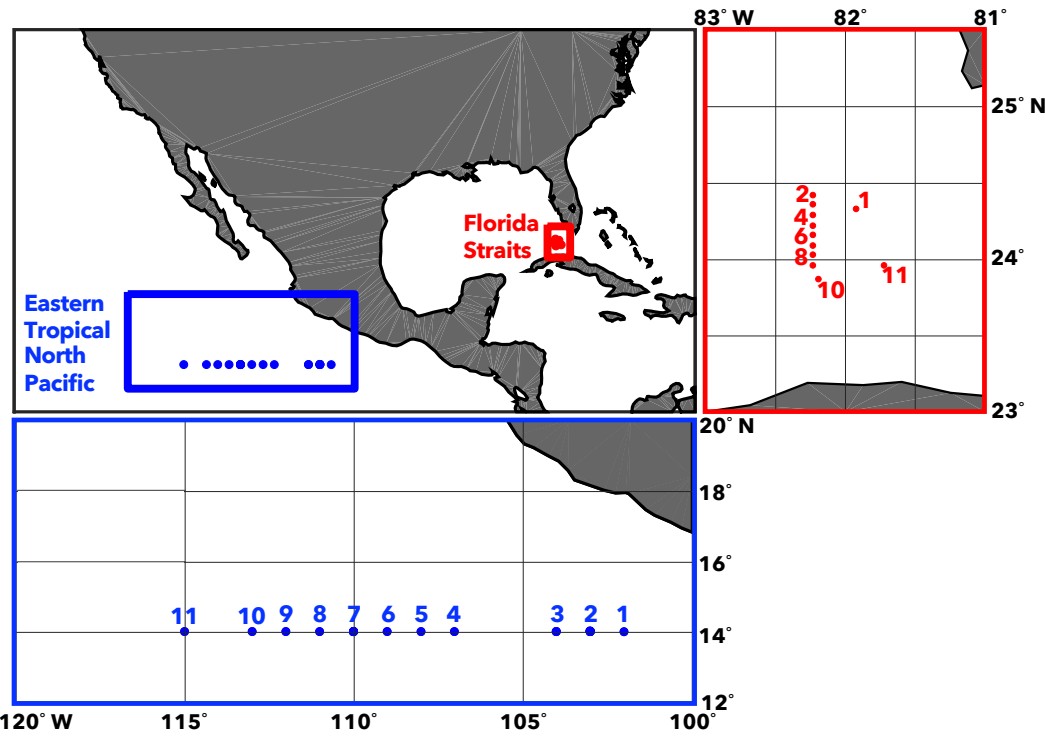

**Figure 1 – Sample Locations.** Locations of 11 samples in both the Eastern Tropical North Pacific (June – July 2018) and the Florida Straits (March 2017). Microlayer and subsurface samples were collected at each location.

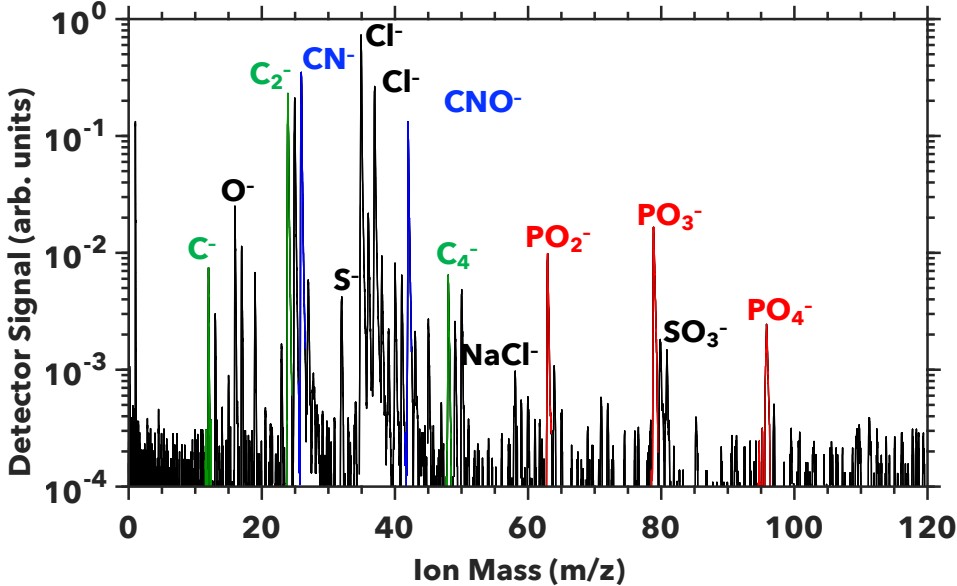

**Figure 2 – SSA Mass Spectrum.** A representative mass spectrum from PALMS (Pacific Microlayer Sample 2) shows indicators of carbon, organic nitrogen, and phosphorus molecules that may enhance ice nucleation activity of SSA (Wolf et al. 2019). Carbon peaks are labeled in green, organic nitrogen in blue, and phosphorus in red. Inorganic peaks are labeled in black.

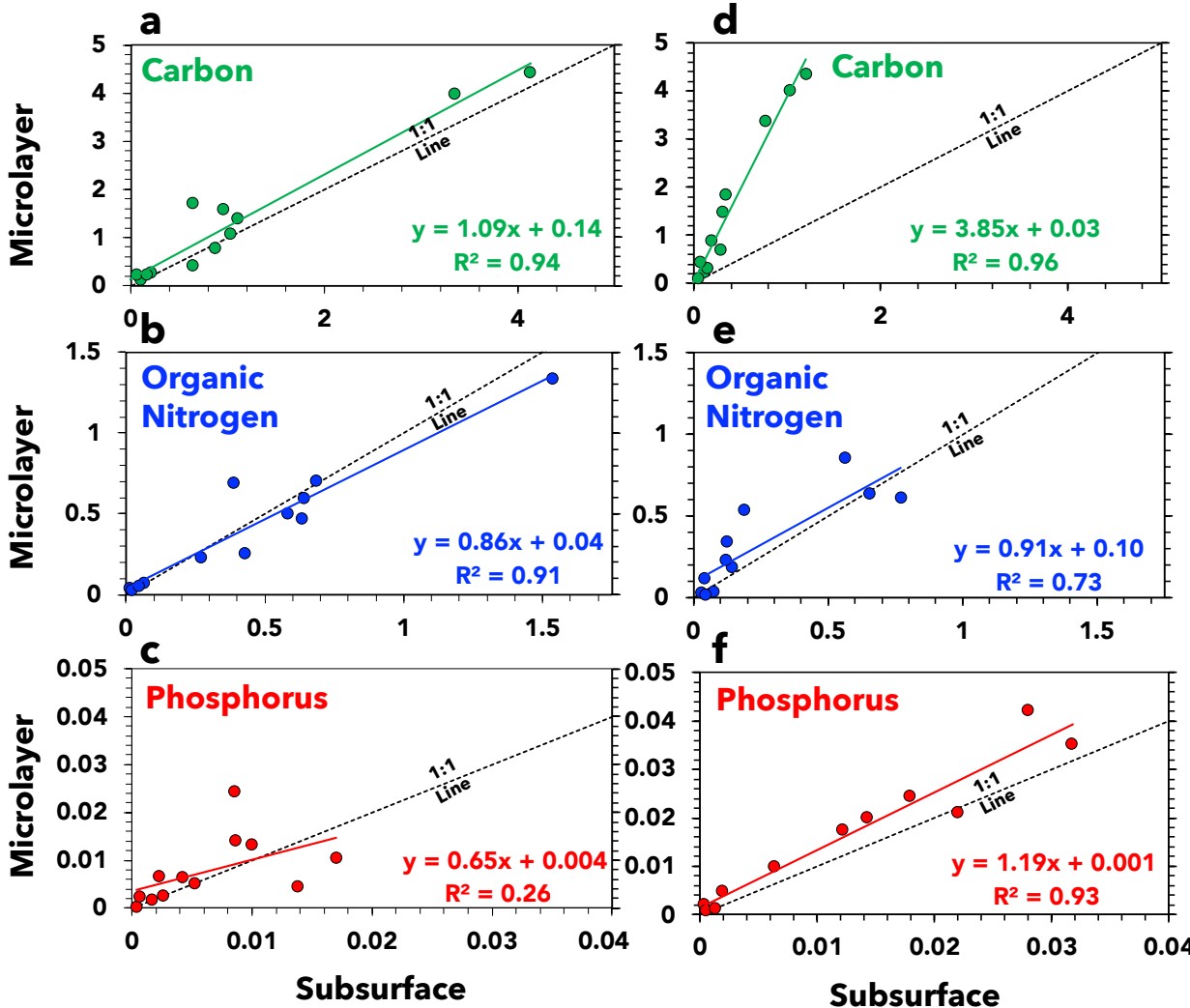

**Figure 3 – Seawater Composition.** The relative abundance of carbon, organic nitrogen, and phosphorus in (**a – c**) ETNP and (**d – f**) Florida Straits seawater samples. Axes represent PALMS ion signals in arbitrary units (Cziczo et al. 2006). Each data point represents the average value of at least one thousand spectra for each sample. Also illustrated are ordinary least squares linear regressions and reference 1:1 lines indicating equal signal in subsurface and microlayer samples.

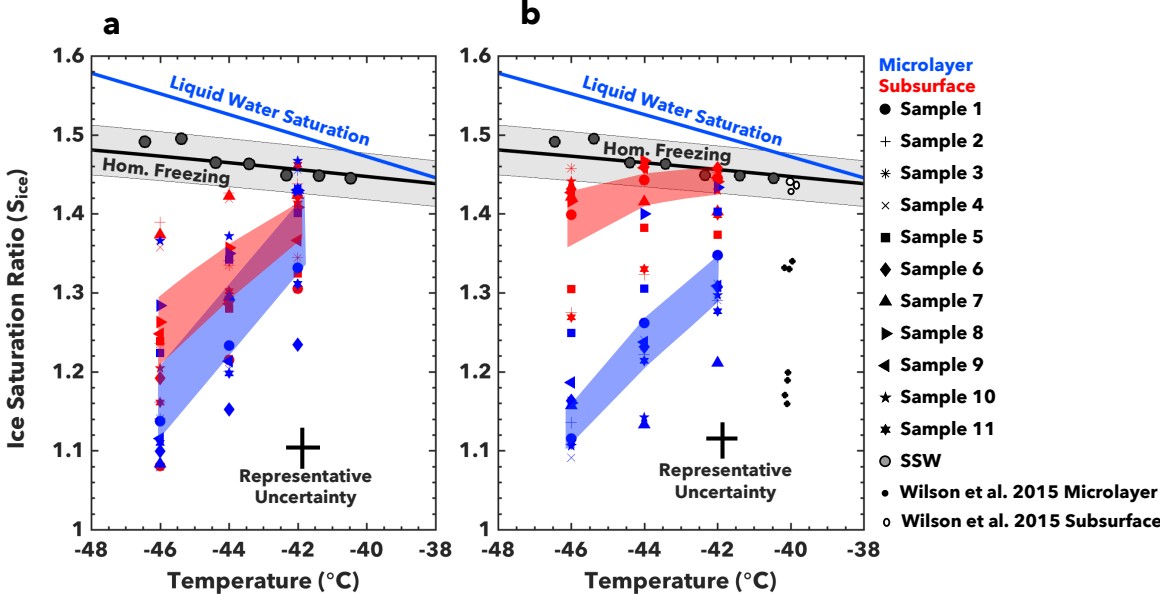

**Figure 4 – Deposition Freezing.** Conditions at the onset of deposition mode ice nucleation are shown for (**a**) ETNP and (**b**) Florida Straits seawater samples. The onset of ice nucleation is defined as when 1% of particles nucleate ice. Data points represent the average critical $S_{ice}$ value at –46, –44, and –42 ˚C. Shaded regions indicate the average critical $S_{ice}$ for all microlayer or subsurface samples, with a standard deviation of variability. Also illustrated are the conditions for homogeneous freezing for particles between 100 and 300 nm in diameter (Koop et al. 2000), as well as the onset of homogeneous nucleation as measured in SPIN with synthetic seawater (SSW) aerosol. Representative uncertainty in onset conditions arises due to variability in SPIN lamina conditions (Garimella et al. 2016).

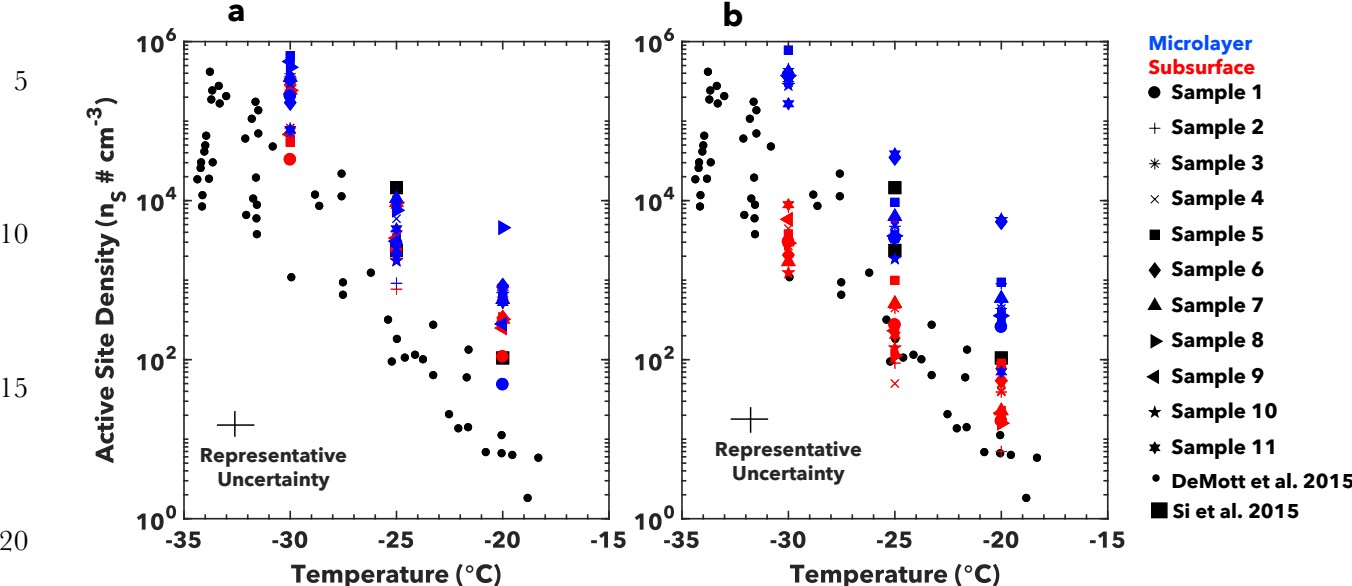

**Figure 5 – Immersion Freezing.** Active site density ($n_s$) for immersion-mode ice nucleation are shown for (**a**) ETNP and (**b**) Florida Straits seawater samples. Data points represent the average value $n_s$ at –30, –25, and –20 ˚C. Representative uncertainties in $n_s$ are derived from variability in replicate experiments, whereas temperature uncertainty arises due to variability in SPIN lamina conditions (Garimella et al. 2016).

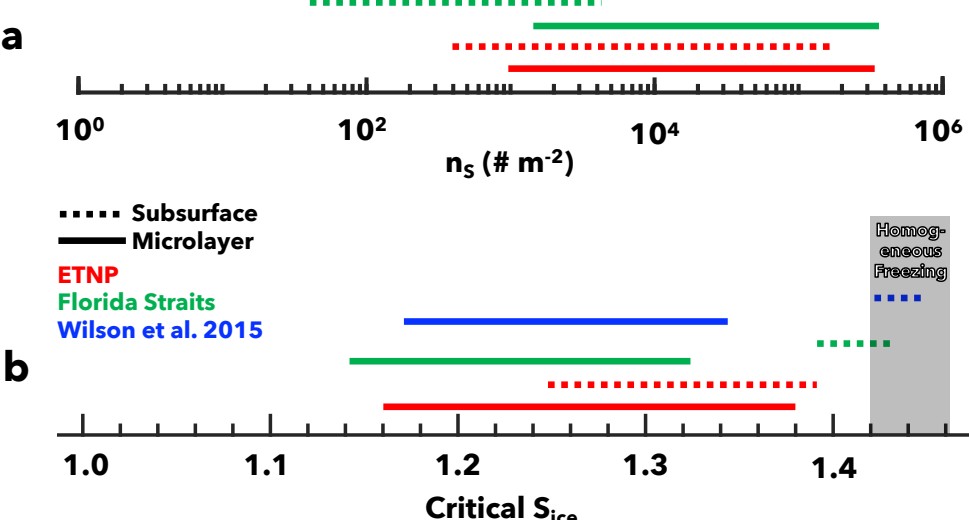

**Figure 6 – Ice Nucleation Activity Comparison.** (a) The INP active site density in the immersion mode, and (b) critical supersaturation required to attain 1% fractional activation in the deposition mode. Subsurface samples from the ETNP were more effective INPs than subsurface samples from less productive marine environments like the Florida Straits and the North Atlantic (Wilson et al. 2015).