# Peer review of "A Link between the Ice Nucleation Activity and the Biogeochemistry of Seawater"

_Atmospheric Chemistry and Physics, 2020_

## Referee Comment (RC1) · Matthew Salter (Referee) · 6 Jul 2020

**1   Overview**

Wolf *et al.* present measurements of the ice-nucleating ability and chemical composition of aerosols generated from sub-surface and sea surface microlayer samples obtained at two locations with contrasting biogeochemistry; the highly productive Eastern Tropical North Pacific Ocean and the less productive Florida Straits. Using this data, the authors present the thesis that "jet droplets aerosolized from the subsurface waters of highly productive regions may therefore be an unrealized source of effective INPs".

[Figure]

Although the dataset is rather limited in scope, I have no reason to doubt the quality of the aerosol composition and ice-nucleation measurements which, presented correctly, may be of interest to ACP readers. However, I have concerns that 1) the manuscript is missing critical information on the methods used to generate the aerosol and 2) the interpretation of the data given the approaches used to generate the aerosol is flawed. As such, my recommendation is that this manuscript should only be accepted following major revisions. Below I outline my concerns as well as more minor points that should be rectified prior to publication in ACP.

**2   Major comments**

1) Seawater aerosolisation - The authors have used an atomizer to generate aerosols which, given the title of the manuscript, they clearly think is representative of nascent sea spray aerosol. There are several major problems with this. Firstly, the authors need to be clear about the drawbacks of using an atomiser to simulate sea spray aerosol and how atomisation differs from the natural bubble bursting process. The size distribution of the aerosol generated by an atomizer will be very different to the size distribution of aerosols generated using both other common laboratory approaches (e.g. laminar and circular plunging jets) and, more critically, natural sea spray aerosol. The chemical composition of the aerosol generated using an atomizer is also going to be very different to the size-dependent composition of nascent sea spray aerosol e.g. O'Dowd et al., 2004 (field evidence) and Prather et al., 2013, Collins et al., 2014 etc. (laboratory evidence). Secondly, atomisation is a very energetic process during which plankton cells may be ruptured allowing ice-nucleating macro-molecules to be dispersed through the aerosol population (e.g. Ickes et al., 2020). The authors should also include some mention of this and that this will once again differentiate the aerosol they generate from that which is formed by bubble bursting at the ocean surface. Thirdly, and most critically, it is completely unacceptable to equate atomisation of sub-surface seawater

samples with jet droplet formation by bubble bursting. As such, all reference to jet droplets in the context of the results and discussion presented by the authors needs to be removed (see relevant lines in the minor comments below).

2) With regards the aerosol generation approach used by the authors, another major issue is that the size distribution of the atomiser used by the authors is not presented anywhere in the manuscript. Indeed the authors also fail to present an adequate description of the "custom" atomiser itself. All of these major issues must be rectified prior to publication in ACP.

3) Critical literature is missing from the introduction - While the introduction is generally well written it is missing a balanced discussion of the dependence of nascent sea spray aerosol composition on seawater biogeochemistry something which is critical given the topic of the manuscript. See my detailed comments below.

**3  Minor comments**

Page 1, Line 31 - "Jet droplets aerosolized from the subsurface waters of highly productive regions may therefore be an unrealized source of effective INPs" should be removed since the authors have not probed jet droplets specifically.

Page 3, Line 21 - The authors do an adequate job of introducing the process of natural sea spray formation in this paragraph. However, they have not introduced the mechanism by which they generate aerosols. This would be an ideal location to contrast the two aerosol formation approaches and the properties of the aerosols that result.

Page 3, Line 27 - The authors state the following: "SSA particles produced from jet drops are composed mainly of inorganic salts but may also contain whole or fragments of cells and soluble organic molecules in subsurface waters (Wilson et al., 2015; Wolf et al., 2019). Film burst particles can contain higher mass fractions of semi-soluble

and insoluble organic molecules in the sea surface microlayer (Cochran et al., 2017)."
While the authors are right to point out the current consensus that film droplets and
jet droplets likely have distinct chemical characteristics I disagree with the use of solubility as a means of distinction. I would argue that there is consensus that it is the
propensity of a molecule to go to the air-sea interface, that is surface-activity, that likely
distinguishes which molecules are more likely to be present in the film droplets than
the jet droplets and that solubility $\neq$ surface-activity when considering the plethora
of organic compounds present in seawater. Two very similar compounds with equal
surface-activity, both of which reduce interfacial free energy, can differ greatly in their
behaviour because of a different degree of bulk solubility. Given this I would suggest
the authors amend this statement.

Page 3, Line 31 - The authors state that "The biogeochemistry of seawater can have
a large impact on the composition of SSA". This is a generalisation that needs to be
expanded upon with reference to the literature. The degree to which the composition
of primary sea spray is affected by biological activity in the surface ocean is a longstanding question in the field. For example, recent field experiments where open ocean
seawater were bubbled indicate that biological productivity has a minor influence on
sea spray organic carbon content and composition (and its CCN properties for that
matter) e.g. Bates et al., 2020; Quinn et al., 2014; Russell et al., 2010. Indeed,
Beaupré et al. (2019) recently reported that highly aged DOM carbon could account for
19-40% of the organic carbon in artificially generated sea spray. In contrast, Ceburnis
et al. (2016) found that most organic enrichment in marine aerosol over the southern
Indian Ocean was attributable to fresh POM. This dichotomy needs to be accurately
represented in the introduction to the manuscript.

Page 4, Line 6 - Since the authors state that " Measurements of INP concentration
and activity from diverse marine regions are relatively rare" they should be able to
provide an overview here. Given this some important recent literature is missing here
(Creaman et al. 2019; McCluskey et al. 2018; Gong et al. 2020; Ickes et al. 2020).

[Figure]

Page 4, Line 14 - "This indicates that jet droplets in these regions may be an overlooked source of INPs" should be removed since the authors have not probed jet droplets specifically.

Page 4, Line 33 and Figure 2 - Sampling using a glass plate is a standard method in use since the early 70's. Given this it has been used in 100's if not 1000's of studies and there is absolutely no need to dedicate a figure in the main manuscript to it. As such, I suggest the authors either completely remove figure 2 or at the very least place it in the supplementary information.

Page 4, Line 34 and Table 1 - The authors state that "rough seas precluded" collection of surface microlayer samples some distance away from the ship. Indeed table 1 shows that the average wind speed at the sampling locations was 15 m s$^{-1}$ and 13.5 m s$^{-1}$ in the Florida straights and the Eastern Tropical North Pacific Ocean, respectively. These are very high wind speeds for sampling surface microlayer (experience tells me this was difficult!). Given this, I think some discussion on the potential impact of such rough seas on both the formation and persistence of the surface microlayer as well as the sampling is warranted here. For example, see the discussion in Rahlff et al. (2017), Sun et al. (2018), Engel et al (2018).

Page 4, Line 37 - Although the authors have used a common approach to estimate the thickness of the sea surface microlayer they sampled, this number is highly uncertain and presenting it suggests higher confidence in it than is warranted. Given that this information is not at all critical to the later discussion I suggest the authors remove the following sentences "Based on the volume of seawater collected per dip and the surface area of the plate, the thickness of the organically-enriched layer adhering to the plate was on average 26 $\mu$m. This falls within the range of previous findings (Irish et al., 2017)."

Page 5, Line 5 - The following issue is certainly not limited to this study but should be mentioned here so that the authors and future readers of this manuscript interested

in conducting similar experiments are aware. Given the high solubility of many of the surfactants enriched at the ocean surface a subsurface sample will rapidly form its on microlayer in a sample bottle or atomiser that may be very similar to a co-located microlayer sample. For example, there is a significant body of literature presenting direct estimates of microlayer formation rates following disruption (e.g. Dragčević and Pravdić, 1981, Kozaraca et al., 2005, Kuznetsova and Lee, 2001, Van-Vleet and Williams, 1983, Williams et al., 1986, Cunliffe et al., 2013) and the current consensus is that they are rapid, typically $< 1$ min. This point further highlights the issue with the authors suggesting atomisation of their sub-surface samples can be equated with jet drop formation.

Page 5, Line 15 - The authors state that they use a "custom Collison-type atomizer" but do not provide any further information. Given the critical role this apparatus has to the study I would like to see either a reference to where it is described in detail or further details here. For instance, a schematic of the atomiser in the supplementary information would be much more useful than a schematic of glass plate sampling.

Page 5, Line 21 - " Particles were size selected (mobility diameter = 200 nm)..." The authors state which size of particles were investigated in terms of the chemical composition and ice-nucleating ability but the reader has no sense of what the overall particle size distribution looked like given that none is presented. If the atomiser the authors used is anything like those I have encountered previously it will produce a narrow size distribution with relatively small particles. However, this is complete speculation until the authors present the size-distribution which they must do.

Page 8, Line 14 - "Our compositional analysis demonstrates that the ocean biogeochemistry impacts the composition of SSA". Given the actual experiments conducted by the authors the language used in this sentence is far too strong. The analysis conducted by the authors demonstrates that aerosols generated by an atomiser from seawater with very different biogeochemical states have differing composition.

Page 9, Line 15 - "This indicates that both jet drop particles originating from subsurface

water and smaller film burst particles originating from the sea surface microlayer in productive marine environments can be effective depositional INPs. These organically-enriched jet droplets can constitute a large fraction of submicrometer SSA (Wang et al., 2017)" should be removed since the authors have not probed jet droplets specifically.

Page 10, Line 12 - "Atomizing seawater creates SSA with more uniform composition than natural seawater aerosolization processes, as it does not mimic the film burst and jet drop aerosolization processes that create organically enriched and depleted SSA, respectively." Here the authors have nicely summarized the major issue with the manuscript in its current form. This discussion belongs much earlier in the manuscript alongside the introduction of the process of film and jet droplet production in natural bubble bursting (see my comments above). Also, I would like to see a reference for the statement "Atomizing seawater creates SSA with more uniform composition than natural seawater aerosolization processes...". Do the authors have evidence for this or is it simply speculation? It is critical when it comes to the next point.

Page 10, Line 16 - "Our derived ns values may therefore be lower estimates for immersion mode INP activity." Following on from my previous point, given that the authors provide no evidence suggesting that atomized seawater has a more "uniform composition than natural seawater aerosolisation processes" this sentence is idle speculation and should be removed. It would be equally unjustified for me to say that the narrow size distribution with small particles that are likely more enriched in organic material compared to larger particles sizes will bias estimated ice nucleation site densities to higher values compared to natural aerosol. Without further information we cannot say either way.

Page 10, Line 34 - "Both film burst and jet droplet particles generated from microlayer and subsurface waters in productive regions such as the ETNP are likely to be sources of effective INPs. In less productive regions, film burst particles may be the dominant source of marine INPs." This statement may well be true but the authors have not generated data that would allow them to test this so both these sentences must be

removed.

Page 11, Line 12 - "The subsurface is aerosolized through bubble bursting mechansism, which create jet droplets (Pruppacher and Klett 1980, Wilson et al. 2015, Wang et al. 2017). This implies that jet droplet aerosols generated in coastal upwelling regions or during spring phytoplankton blooms can be a source of INPs." Again, the authors have not generated data that would allow them to test this so both these sentences must be removed.

Page 11, Line 34 - "However, our results demonstrate that larger jet drop particles originating from highly productive subsurface waters may be a source of effective INPs as well." Again, the authors have not generated data that would allow them to test this so both these sentences must be removed.

**4  References**

Bates, T. S., Quinn, P. K., Coffman, D. J., Johnson, J. E., Upchurch, L., Saliba, G., ... Behrenfeld, M. J. (2020). Variability in Marine Plankton Ecosystems Are Not Observed in Freshly Emitted Sea Spray Aerosol Over the North Atlantic Ocean. Geophysical Research Letters, 47(1), e2019GL085938.

Beaupré, S. R., Kieber, D. J., Keene, W. C., Long, M. S., Maben, J. R., Lu, X., ... Chang, R. Y. W. (2019). Oceanic efflux of ancient marine dissolved organic carbon in primary marine aerosol. Science advances, 5(10), eaax6535.

Ceburnis, D., Masalaite, A., Ovadnevaite, J., Garbaras, A., Remeikis, V., Maenhaut, W., ... O'Dowd, C. D. (2016). Stable isotopes measurements reveal dual carbon pools contributing to organic matter enrichment in marine aerosol. Scientific reports, 6(1), 1-6.

Creamean, J. M., J. N. Cross, R. Pickart, L. McRaven, P. Lin, A. Pacini, R. Hanlon, D.

G. Schmale, J. Ceniceros, T. Aydell, N. Colombi, E. Bolger, and P. J. DeMott (2019), Ice Nucleating Particles Carried From Below a Phytoplankton Bloom to the Arctic Atmosphere, Geophys. Res. Lett., 46(14), 8572-8581, doi:10.1029/2019gl083039.

Cunliffe, M., Engel, A., Frka, S., Gašparović, B., Guitart, C., Murrell, J. C., ... Wurl, O. (2013). Sea surface microlayers: A unified physicochemical and biological perspective of the air–ocean interface. Progress in Oceanography, 109, 104-116.

Dragcevic, D., Pravdic, V. (1981). Properties of the seawater‐air interface. 2. Rates of surface film formation under steady state conditions 1. Limnology and Oceanography, 26(3), 492-499.

Engel, A., Sperling, M., Sun, C., Grosse, J., Friedrichs, G. (2018). Organic matter in the surface microlayer: insights from a wind wave channel experiment. Frontiers in Marine Science, 5, 182.

Gong, X., H. Wex, M. van Pinxteren, N. Triesch, K. W. Fomba, J. Lubitz, C. Stolle, B. Robinson, T. Müller, H. Herrmann, and F. Stratmann (2020), Characterization of aerosol particles at Cape Verde close to sea and cloud level heights - Part 2: ice nucleating particles in air, cloud and seawater, Atmos. Chem. Phys., 20, 1451-1468, doi:10.5194/acp-20-1451-2020.

Ickes, L., Porter, G. C. E., Wagner, R., Adams, M. P., Bierbauer, S., Bertram, A. K., Bilde, M., Christiansen, S., Ekman, A. M. L., Gorokhova, E., Höhler, K., Kiselev, A. A., Leck, C., Möhler, O., Murray, B. J., Schiebel, T., Ullrich, R., and Salter, M.: Arctic marine ice nucleating aerosol: a laboratory study of microlayer samples and algal cultures, Atmos. Chem. Phys. Discuss., https://doi.org/10.5194/acp-2020-246

Kozarac, Z., Risović, D., Frka, S., Möbius, D. (2005). Reflection of light from the air/water interface covered with sea-surface microlayers. Marine chemistry, 96(1-2), 99-113.

Kuznetsova, M., Lee, C. (2001). Enhanced extracellular enzymatic peptide hydrolysis

in the sea-surface microlayer. Marine Chemistry, 73(3-4), 319-332.

McCluskey, C. S., J. Ovadnevaite, M. Rinaldi, J. Atkinson, F. Belosi, D. Ceburnis, S. Marullo, T. C. J. Hill, U. Lohmann, Z. A. Kanji, C. O'Dowd, S. M. Kreidenweis, and P. J. DeMott (2018), Marine and Terrestrial Organic Ice-Nucleating Particles in Pristine Marine to Continentally Influenced Northeast Atlantic Air Masses, J. Geophys. Res.-Atmos., 123(11), 6196-6212, doi:10.1029/2017jd028033

O'Dowd, C. D., Facchini, M. C., Cavalli, F., Ceburnis, D., Mircea, M., Decesari, S., et al. (2004). Biogenically driven organic contribution to marine aerosol. Nature, 431(7009), 676–680. https://doi.org/10.1038/nature02959

Prather, K. A., Bertram, T. H., Grassian, V. H., Deane, G. B., Stokes, M. D., DeMott, P. J., et al. (2013). Bringing the ocean into the laboratory to probe the chemical complexity of sea spray aerosol. Proceedings of the National Academy of Sciences, 110(19), 7550–7555. https://doi.org/10.1073/pnas.1300262110https://www.frontiersin.org/articles/10.3389/fmars.2018.00182/full

Quinn, P. K., Bates, T. S., Schulz, K. S., Coffman, D. J., Frossard, A. A., Russell, L. M., ... Kieber, D. J. (2014). Contribution of sea surface carbon pool to organic matter enrichment in sea spray aerosol. Nature Geoscience, 7(3), 228-232.

Rahlff J, Stolle C, Giebel HA, et al. High wind speeds prevent formation of a distinct bacterioneuston community in the sea-surface microlayer. FEMS Microbiol Ecol. 2017;93(5):fix041. doi:10.1093/femsec/fix041

Russell, L. M., Hawkins, L. N., Frossard, A. A., Quinn, P. K., Bates, T. S. (2010). Carbohydrate-like composition of submicron atmospheric particles and their production from ocean bubble bursting. Proceedings of the National Academy of Sciences, 107(15), 6652-6657.

Sun, C.-C., Sperling, M., and Engel, A.: Effect of wind speed on the size distribution of gel particles in the sea surface microlayer: insights from a wind–wave channel experiment, Biogeosciences, 15, 3577–3589, https://doi.org/10.5194/bg-15-3577-2018, 2018.

Van Vleet, E. S., Williams, P. M. (1983). Surface potential and film pressure measurements in seawater systems 1. Limnology and Oceanography, 28(3), 401-414.

Williams, P. M., Carlucci, A. F., Henrichs, S. M., Van Vleet, E. S., Horrigan, S. G., Reid, F. M. H., Robertson, K. J. (1986). Chemical and microbiological studies of sea-surface films in the Southern Gulf of California and off the West Coast of Baja California. Marine Chemistry, 19(1), 17-98.

---

## Referee Comment (RC2) · Anonymous Referee #1 · 23 Jul 2020

Review of "A Link between the Ice Nucleation Activity of Sea Spray Aerosol and the Biogeochemistry of Seawater." by Wolf et al., submitted to ACPD

The study examines laboratory generated particles from samples of seawater and the surface microlayer from two different locations wrt. their ice nucleation ability. It is an interesting study, showing that oceanic productivity and ice nucleation ability of the related particles are somewhat connected. It is suggested that jet droplets occurring during sea spray production might play a larger role for atmospheric INP, which, however, is not really examined in the study itself, as all examined particles are generated artificially and the sea spray particle generation mechanism was not examined at all.

While the topic of the paper is interesting, writing needs to be improved in a number of locations. Particularly the introduction needs to be improved a lot. It does not really focus on marine INP (which it should have), but instead is a broad collection of information given in detail which would not need to be so detailed (such as different types of INP or mixed phase and cirrus clouds). This contrasts with the fact that publications dealing with topics related to the content of this manuscript are missing. The focus of this introduction does not fit to the scope of the manuscript. Comments on the "Introduction"-section are therefore given separately below.

Another more general concern is the use of the word SSA (sea spray aerosol) for the particles examined here. SSA is a specific aerosol generated by wave activity and bubble bursting - and then an aerosol always includes particles as well as the gas-phase around them. Strictly speaking, the study examines particles generated from sea water samples. To avoid confusion, I would recommend using SWP (sea water particles) or such. Also, it needs to be check throughout the text if it is referred to particles or really all of the aerosol. Generally, the use of "P" (particle) instead of "A" (aerosol) will be better.

Once these issues, together with the other more detailed ones below will have been addressed, the manuscript can be considered for publication in ACP. But a thorough revision of the manuscript is needed at first.

Introduction:

p2, lines 8-11: These two sentences (starting with "Ice formation" and ending with "important") don't make sense together. Ice-formation (meaning the mechanism of ice nucleation) is NOT the Wegener-Bergeron-Findeisen effect. The latter concerns growth of ice crystals even in regions that have relative humidities < 100% wrt. liquid water. These sentences need to be completely reformulated.

p2, line 17: Citing Whale et al. (2018) for this is awkward as this is textbook knowledge.

p2, lines 17-30: The different heterogeneous ice nucleation mechanisms are not clearly described, and their importance is not mentioned (immersion freezing is thought to be the most important for mixed phase clouds, for cirrus this is not clear yet). Instead, remarks are made on comparably unimportant effects. This needs to be rewritten.

p2, line 20: "pore condensation and freezing" was first suggested and examined in Marcolli (2014), so it would be fair to cite that publication here. Or to skip that mentioning completely, as this is not what you are looking at.

p2, lines 33-37: There is no need to go into so much detail for types of INP which are certainly NOT emitted by the ocean. It is also somewhat unclear which citation here is given for which type of INP. Also, there are good reviews which you could cite instead, two of which you already used above (Hoose & Möhler 2012; Murray et al., 2012), but also a much older one (Szyrmer and Zawadzki, 1997) and a newer one (Kanji et al., 2017) - it would be better to cite reviews here instead of your selection, which often does not include the oldest / newest / most cited publication for the separate INP types, anyway, and which is too detailed, given your focus on marine INP.

p3, line 1: You miss all the new work on that, which should not have happened, given that this is the topic you are focusing on in here. It's weird that here now you cite review papers, on the topic you want to look at in depth. Just a selection: Burrows et al. (2013), Creamean et al. (2019), Gong et al. (2020), Ladino et al. (2019), McCluskey et al. (2018a,b).

p3, line 8: Also the link from the ocean to the atmosphere is important for your claim, and that is understudies, too! Particularly three recent publications (already included above) might be important in this respect, as they are dealing with marine INP (which necessarily includes sea spray): McCluskey et al. (2018a,b) find low INP concentrations in remote marine regions (Southern Ocean and North East Atlantic), Gong et al. (2020) find that marine INP contribute only a very small fraction of atmospheric INP in Cape Verde. As these publications are directly linked to your topic they should be discussed in your work. Also the above mentioned publication by Creamean et al. (2019) might be of interest in that respect.

p3, line 9: "DeMott et al. (2015)": I guess you mean the one that is given as "(2015b)" in your references? But that actually is "(2016)", anyway. (The preprint came out shortly before new years in 2015, but the final printing date was in 2016).

p3, lines 14-16: Is the content of this sentence related to Wilson et al. (2015) (which you cite in the beginning of the sentence before) or to Knopf et al. (2011) (which you cite at the end of the next sentence)? Clarify!

p3, line 21: "SSA encompasses a range of particle chemistries": SSA does not "encompass" particle chemistries. The chemistry goes on in the SML or sea water. Reformulate! Or if you want to point to the next sentence, then it would be "The formation of SSA encompasses a range of physical processes that affect / are affected ... ."

General comments:

p5, line 15: Give more information on the working principle of the atomizer. Your readers need to know how the particles were generated. The generation process is a big part of atmospheric SSA, in terms of particle sizes, particle concentrations and particle composition, and this is not easily reproduced with just generating particles from sea water or SML samples. You can check for some information on this issue in the introduction of Fuentes et al. (2010). "Real-world-line" SSA likely is best obtained by using wave channels (Prather et al., 2013). The generation technique you use is rather just a means to generate particles, but if they are similar to atmospheric particles generated from sea spray is a separate issue.

p5, line 21: Choosing a mobility diameter of 200nm implies that you assume that marine INP are all separately floating (likely biogenic) macromolecules. Mention that explicitly, and elaborate on that - that is one thing that could also be discussed already in the introduction. You need to justify why you can assume that this choice will not cause

you to lose the majority of all INP (see Wilson et al. (2015) and Irish et al. (2017), which you already cite).

p5, lines 22-24: Doubly and triply changed 200nm particles should still be smaller than 500nm, so the choice of your cut-off might not have been optimal. You can still argue that this will remove the more highly charged particles (which, however, do not occur in such high amounts). Please check this, and also correct the text accordingly, so that others will not copy this (wrong) approach in the future. An estimation on the influence of multiply charged particles on the final results is needed.

p7, line 6: This was mentioned above, but again (and valid for the whole text): The use of "SSA" for artificially generated aerosol is a bit misleading.

p7, lines 19-26: Check this whole paragraph, together with Fig. 4. There are inconsistencies! I think that the following could remove these issues, but please check carefully for yourself: (a) line 21 -> change 4d to 4a! (b) line 25 -> change 4a to 4d! (c) "This suggests that subsurface waters in the ETNP supported more primary production." - I assume you mean compared to the Florida Straits? Then please say so explicitly!

p7, lines 28-29: Comparing averages here might seem the wrong choice, as the higher values for ETNP come from two outliers (at least to some extent). However, also if you gave the median values, your statement would still be correct (as far as I can see). Therefore, it might be worth adding the median values as well.

p8, lines 5-6: "PALMS 5 detecting ionization of more soluble organic nitrogen species in addition to amino acids." - And you assume that one would be present in higher concentration in the subsurface, the other one more in the microlayer water? Clarify! Also: Which type of detection was used by Zäncker (previous sentence)? And are you aware of the following publications: Kuznetsova & Lee (2002), Kuznetsova et al. (2004), Reinthaler et al. (2008) and Engel & Galgani (2016), who all found amino acids enriched in the surface microlayer in different oceanic regions. This may support your hypothesis, that the ionization efficiency of different compounds in PALMS might influence your results, which, however, then would have an influence on the interpretation of your data. Enrichment in the surface microlayer is not my main expertise, so take this merely as a suggestion for something you could look into.

p8, line 15: "similar and elevated organic contents" - well, this is not true for the microlayer, which has quite similar values for ETNP and Florida Straits. Only the subsurface value is higher for ETNP than for Florida Straits.

p8, lines 17-18: This last sentence of this paragraph might not be true for INP. As we don't know what it is that makes the INP, this conclusion cannot be drawn! Remove!

p9, lines 5-7: Zeppenfeld et al. (2019) finds a connection between glucose and INP concentrations in sea water samples, for real world measurements on Arctic samples. Please add.

p9, lines 17-18: I am not sure if all readers will be familiar with the study by Wang et al. (2017) which you relate to here. As this is very important for this paragraph and also for conclusions you make later, explicitly describe somewhere in your text (maybe in this paragraph here), that Wang et al. (2017) claims that jet droplets can make up a substantial fraction of all sea spray generated particles. It would be good to also be more specific instead of just writing "a large fraction".

p10, equation 1: This is an approximation which only works for really small fi ($\sim$ up to 0.1, as they are typically measured for example in the AIDA cloud chamber - but I guess you might have had higher fractions in SPIN). Otherwise, the full exponential has to be used!

p10, lines 32-33: Which surface area did DeMott use? (Hint: it is different from how you did it.) You should say explicitly in your study that the total marine aerosol includes a lot of other particles and that the n_s value you give here cannot simply be used for upscaling!

p11, lines 4-5: Here you can see very clearly that my earlier suggestion to use another

name for the particles you look at, instead of SSA, makes sense. An aerosol cannot be an INP - a particle can be an INP. So in any case, "aerosols" in line 5 should become "particles". Please check the whole text to see if you really refer to aerosol only where you mean the combination of particles with the surrounding air!

p11, lines 22-24: And yes, not only modeling studies show that, but also some of those publications using real world data which I suggested above. This should be added here.

p11, lines 29-30: "... (McCluskey et al., 2018). This demonstrates that even highly productive marine environments are not always effective sources of INPs." This is misleading - it depends on the interpretation of "effective". If no (typically strong) land sources are nearby, then the marine environment is the only remaining source, and therefore it could be "effective". Reformulate!

Specific comments:

p2, line 14: last word "from" -> should be "form" p3, line 24: "eject" -> "ejects" p3, line 28: Shouldn't "in" be "included in" or "from"? p7, line 8: Following "signals", add "with PALMS". p8, line 2: "2017" -> "(2017)" p8, line 5: "(Figures 4b and 4e)" should only be "(Figure 4e)", or you should also mention ETNP in this sentence. p9, line 9: "2015" -> "(2015)" p11, line 18: "that that" - remove one of them.

Literature:

Burrows, S. M., C. Hoose, U. Poeschl, and M. G. Lawrence (2013), Ice nuclei in marine air: biogenic particles or dust?, Atmos. Chem. Phys., 13(1), 245-267, doi:10.5194/acp-13-245-2013.

Creamean, J. M., J. N. Cross, R. Pickart, L. McRaven, P. Lin, A. Pacini, R. Hanlon, D. G. Schmale, J. Ceniceros, T. Aydell, N. Colombi, E. Bolger, and P. J. DeMott (2019), Ice Nucleating Particles Carried From Below a Phytoplankton Bloom to the Arctic Atmosphere, Geophys. Res. Lett., 46(14), 8572-8581, doi:10.1029/2019gl083039.

[Figure]

Engel, A., and L. Galgani (2016), The organic sea-surface microlayer in the upwelling region off the coast of Peru and potential implications for air-sea exchange processes, Biogeosciences, 13(4), 989-1007, doi:10.5194/bg-13-989-2016.

Fuentes, E., H. Coe, D. Green, G. De Leeuw, and G. McFiggans (2010), Laboratory-generated primary marine aerosol via bubble-bursting and atomization, Aerosol Measurement Techniques, 3, 141-162.

Gong, X., H. Wex, M. van Pinxteren, N. Triesch, K. W. Fomba, J. Lubitz, C. Stolle, B. Robinson, T. Müller, H. Herrmann, and F. Stratmann (2020), Characterization of aerosol particles at Cape Verde close to sea and cloud level heights - Part 2: ice nucleating particles in air, cloud and seawater, Atmos. Chem. Phys., 20, 1451-1468, doi:10.5194/acp-20-1451-2020.

Irish, V. E., P. Elizondo, J. Chen, C. Chou, J. Charette, M. Lizotte, L. A. Ladino, T. W. Wilson, M. Gosselin, B. J. Murray, E. Polishchuk, J. P. D. Abbatt, L. A. Miller, and A. K. Bertram (2017), Ice-nucleating particles in Canadian Arctic sea-surface microlayer and bulk seawater, Atmos. Chem. Phys., 17(17), 10583-10595, doi:10.5194/acp-17-10583-2017.

Kanji, Z. A., L. A. Ladino, H. Wex, Y. Boose, M. Kohn, D. Cziczo, and M. Krämer (2017), Chapter 1: Overview of Ice Nucleating Particles, in Ice Formation and Evolution in Clouds and Precipitation: Measurement and Modeling Challenges, edited, Meteor. Monogr., doi:10.1175/AMSMONOGRAPHS-D-16-0006.1.

Knopf, D. A., P. A. Alpert, B. Wang, and J. Y. Aller (2011), Stimulation of ice nucleation by marine diatoms, Nat. Geosci., 4(2), 88-90, doi:10.1038/ngeo1037. Kuznetsova, M., and C. Lee (2002), Dissolved free and combined amino acids in nearshore seawater, sea surface microlayers and foams: Influence of extracellular hydrolysis, Aquatic Sciences, 64(3), 252-268, doi:10.1007/s00027-002-8070-0.

Kuznetsova, M., and C. Lee (2002), Dissolved free and combined amino acids in

nearshore seawater, sea surface microlayers and foams: Influence of extracellular hydrolysis, Aquatic Sciences, 64(3), 252-268, doi:10.1007/s00027-002-8070-0.

Kuznetsova, M., C. Lee, J. Aller, and N. Frew (2004), Enrichment of amino acids in the sea surface microlayer at coastal and open ocean sites in the North Atlantic Ocean, Limnology and Oceanography, 49(5), 1605-1619, doi:10.4319/lo.2004.49.5.1605.

Ladino, L. A., G. B. Raga, H. Alvarez-Ospina, M. A. Andino-Enríquez, I. Rosas, L. Martínez, E. Salinas, J. Miranda, Z. Ramírez-Díaz, B. Figueroa, C. Chou, A. K. Bertram, E. T. Quintana, L. A. Maldonado, A. García-Reynoso, M. Si, and V. E. Irish (2019), Ice-nucleating particles in a coastal tropical site, Atmos. Chem. Phys., 19(9), 6147-6165, doi:10.5194/acp-19-6147-2019.

Marcolli, C. (2014), Deposition nucleation viewed as homogeneous or immersion freezing in pores and cavities, Atmos. Chem. Phys., 14(4), 2071-2104, doi:10.5194/acp-14-2071-2014.

McCluskey, C. S., T. C. J. Hill, R. S. Humphries, A. M. Rauker, S. Moreau, P. G. Strutton, S. D. Chambers, A. G. Williams, I. McRobert, J. Ward, M. D. Keywood, J. Harnwell, W. Ponsonby, Z. M. Loh, P. B. Krummel, A. Protat, S. M. Kreidenweis, and P. J. DeMott (2018a), Observations of Ice Nucleating Particles Over Southern Ocean Waters, Geophys. Res. Lett., 45(21), 11989-11997, doi:10.1029/2018gl079981.

McCluskey, C. S., J. Ovadnevaite, M. Rinaldi, J. Atkinson, F. Belosi, D. Ceburnis, S. Marullo, T. C. J. Hill, U. Lohmann, Z. A. Kanji, C. O'Dowd, S. M. Kreidenweis, and P. J. DeMott (2018b), Marine and Terrestrial Organic Ice-Nucleating Particles in Pristine Marine to Continentally Influenced Northeast Atlantic Air Masses, J. Geophys. Res.-Atmos., 123(11), 6196-6212, doi:10.1029/2017jd028033.

Prather, K. A., T. H. Bertram, V. H. Grassian, G. B. Deane, M. D. Stokes, P. J. DeMott, L. I. Aluwihare, B. P. Palenik, F. Azam, J. H. Seinfeld, R. C. Moffet, M. J. Molina, C. D. Cappa, F. M. Geiger, G. C. Roberts, L. M. Russell, A. P. Ault, J. Baltrusaitis, D. B.

Collins, C. E. Corrigan, L. A. Cuadra-Rodriguez, C. J. Ebben, S. D. Forestieri, T. L. Guasco, S. P. Hersey, M. J. Kim, W. F. Lambert, R. L. Modini, W. Mui, B. E. Pedler, M. J. Ruppel, O. S. Ryder, N. G. Schoepp, R. C. Sullivan, and D. Zhao (2013), Bringing the ocean into the laboratory to probe the chemical complexity of sea spray aerosol, Proc. Natl. Acad. Sci. USA, 110(19), 7550-7555, doi:10.1073/pnas.1300262110.

Reinthaler, T., E. Sintes, and G. J. Herndl (2008), Dissolved organic matter and bacterial production and respiration in the sea-surface microlayer of the open Atlantic and the western Mediterranean Sea, Limnology and Oceanography, 53(1), 122-136, doi:10.4319/lo.2008.53.1.0122. Szyrmer, W., and I. Zawadzki (1997), Biogenic and anthropogenic sources of ice-forming nuclei: A review, BAMS, 78(2), 209-228.

Wang, X., G. B. Deane, K. A. Moore, O. S. Ryder, M. D. Stokes, C. M. Beall, D. B. Collins, M. V. Santander, S. M. Burrows, C. M. Sultana, and K. A. Prather (2017), The role of jet and film drops in controlling the mixing state of submicron sea spray aerosol particles, 114(27), 6978-6983, doi:10.1073/pnas.1702420114.

Wilson, T. W., L. A. Ladino, P. A. Alpert, M. N. Breckels, I. M. Brooks, J. Browse, S. M. Burrows, K. S. Carslaw, J. A. Huffman, C. Judd, W. P. Kilthau, R. H. Mason, G. McFiggans, L. A. Miller, J. J. Najera, E. Polishchuk, S. Rae, C. L. Schiller, M. Si, J. V. Temprado, T. F. Whale, J. P. S. Wong, O. Wurl, J. D. Yakobi-Hancock, J. P. D. Abbatt, J. Y. Aller, A. K. Bertram, D. A. Knopf, and B. J. Murray (2015), A marine biogenic source of atmospheric ice-nucleating particles, Nature, 525(7568), 234-238, doi:10.1038/nature14986.

Zeppenfeld, S., M. van Pinxteren, M. Hartmann, A. Bracher, F. Stratmann, and H. Herrmann (2019), Glucose as a potential chemical marker for ice nucleating activity in Arctic seawater and melt pond samples, Environmental Science & Technology, 53(15), 8747-8756, doi:10.1021/acs.est.9b01469.

---

## Author Comment (AC1) · 15 Sep 2020

Dear Editor,

We would like to thank the reviewers for their careful reading of this manuscript and their suggestions to improve it. In response to reviewer comments, we have made several substantial revisions to the text. We believe the new manuscript addresses reviewer concerns and as a result is scientifically improved.

Our point-by-point response to reviewer comments is included below. For clarity, reviewer comments are in green text and our responses are in black. Our line numbers reference the updated document, not the original.

**Reviewer 1**

Wolf et al. present measurements of the ice-nucleating ability and chemical composition of aerosols generated from sub-surface and sea surface microlayer samples obtained at two locations with contrasting biogeochemistry; the highly productive Eastern Tropical North Pacific Ocean and the less productive Florida Straits. Using this data, the authors present the thesis that "jet droplets aerosolized from the subsurface waters of highly productive regions may therefore be an unrealized source of effective INPs".

Although the dataset is rather limited in scope, I have no reason to doubt the quality of the aerosol composition and ice-nucleation measurements which, presented correctly, may be of interest to ACP readers.

We thank the reviewer for carefully considering our manuscript and for reiterating the potential value of our data to readers of ACP.

However, I have concerns that 1) the manuscript is missing critical information on the methods used to generate the aerosol and 2) the interpretation of the data given the approaches used to generate the aerosol is flawed. As such, my recommendation is that this manuscript should only be accepted following major revisions. Below I outline my concerns as well as more minor points that should be rectified prior to publication in ACP.

The reviewer raises several points here which we address in detail below. Briefly, the reviewer highlights our oversight to include a rigorous description of our aerosolization technique using an atomizer. The reviewer also voices concerns about the interpretation of our results. We have made several substantial modifications to the manuscript to address these concerns, including an expanded methodology section as well as removing all claims that we have accurately recreated ambient sea spray aerosol using our laboratory aerosolization technique. We believe the manuscript is scientifically improved as a result of these suggestions.

1. Seawater aerosolisation - The authors have used an atomizer to generate aerosols which, given the title of the manuscript, they clearly think is representative of nascent sea spray aerosol. There are several major problems with this.

We thank the reviewer for clearly articulating their concerns. Below, we address three issues raised by the reviewer and highlight specific changes made to the manuscript in response.

    a. Firstly, the authors need to be clear about the drawbacks of using an atomiser to simulate sea spray aerosol and how atomisation differs from the natural bubble bursting process. The size distribution of the aerosol generated by an atomizer will be very different to the size distribution of aerosols

generated using both other common laboratory approaches (e.g. laminar and circular plunging jets) and, more critically, natural sea spray aerosol. The chemical composition of the aerosol generated using an atomizer is also going to be very different to the size-dependent composition of nascent sea spray aerosol e.g. O'Dowd et al.,2004 (field evidence) and Prather et al., 2013, Collins et al., 2014 etc. (laboratory evidence).

We have clarified in our methodology section that the atomization technique – although previously employed by recent publications – has major shortcomings with regards to the size distribution and composition of the resulting aerosol. Specifically, we have added the following paragraph of text:

> We note that the atomization technique – although used in prior studies investigating the ice nucleation of sea spray aerosol (Ladino et al., 2016; Wilson et al., 2015) – has several limitations. Specifically, atomization produces aerosol with different physical and chemical characteristics than ambient SSA (Collins et al., 2014). First, atomization results in a different aerosol size distribution due to the lack of bubble bursting mechanisms (Fuentes et al., 2010). The impact of this artefact can be limited by size-selecting a narrow diameter range from the resulting polydisperse aerosol stream prior to INP analyses. However, atomization also produces aerosols of a different composition than ambient SSA (Gaston et al., 2011). Natural bubble-bursting mechanisms result in aerosol with size-dependent composition ((Collins et al., 2014; O'Dowd et al., 2004; Prather et al., 2013). It is unlikely that atomization can replicate the composition of natural SSA. (Page 6 Lines 21-29)

We thank the reviewer for recommending several references, which we have included (among others) in the text.

b.  Secondly, atomisation is a very energetic process during which plankton cells may be ruptured allowing ice-nucleating macro-molecules to be dispersed through the aerosol population (e.g. Ickes et al., 2020). The authors should also include some mention of this and that this will once again differentiate the aerosol they generate from that which is formed by bubble bursting at the ocean surface.

The reviewer raises a valid point here which further highlights the shortcomings of the atomization technique. We now refer to the energetic nature of the atomization process in two places in the text:

> Further, atomization is an energetic process that may result in a higher rate of cell lysis than expected from natural processes, such as apoptosis, viral infection, or predator grazing (Agustí and Duarte, 2013; Kirchman, 1999). This may artificially increase the organic content of our laboratory-generated aerosol and increase the occurrence of ice nucleating macromolecules in particles (Ickes et al., 2020; Knopf et al., 2011). (Page 6 Lines 29-32)

> We note that the atomization process energetically aerosolizes the seawater solution, potentially rupturing cells, resulting in particles with more INP-active organic macromolecules than might occur in natural SSA. (Page 12 Lines 16-18)

c.  Thirdly, and most critically, it is completely unacceptable to equate atomisation of sub-surface seawater samples with jet droplet formation by bubble bursting. As such, all reference to jet droplets in the context of the results and discussion presented by the authors needs to be removed (see relevant lines in the minor comments below).

We thank the reviewer for identifying instances where we have made inaccurate claims. We have made several revisions to the text to address this concern, as detailed in our response to the specific

comments below. We no longer equate or draw parallels between ambient jet-droplets and atomization of subsurface seawater.

2. With regards the aerosol generation approach used by the authors, another major issue is that the size distribution of the atomiser used by the authors is not presented anywhere in the manuscript. Indeed the authors also fail to present an adequate description of the "custom" atomiser itself. All of these major issues must be rectified prior to publication in ACP.

The reviewer raises several points here, which are then reiterated in the specific comments below. Here we provide a broad overview of changes made to the manuscript in response to these points.

First, the reviewer requests that we include a size distribution of the aerosol particles resulting from our atomizer. We have now included a size distribution in the Supplemental Information section. However, we emphasize that our INP and experiments did not use polydisperse aerosol particles, but rather 200 nm diameter particles size-selected using a Differential Mobility Analyzer after atomization. We have clarified this in several places in the text:

> The resulting dried sea salt aerosols were diverted into a differential mobility analyzer (DMA, Model 2002; Brechtel Manufacturing Inc., Hayward, CA). *Particles were size selected (mobility diameter = 200 nm*) with a sheath to sample flow ratio of 8:1. (Page 7 Lines 4-6)

> We investigated the composition of *200 nm* SSA generated from seawater samples using the Particle Analysis by Laser Mass Spectrometry (PALMS) instrument (Cziczo et al., 2006). (Page 7 Lines 22-23).

> *Size-selected aerosol particles* were drawn into the nucleation chamber and nominally constrained to a flow centerline with particle free sheath air adjacent to the ice-covered walls. (Page 8 Lines 7-8).

We further indicate the size selection in our new SI Figure with a vertical line indicating the diameter of size selection (200 nm).

Second, the reviewer requests we provide more detail on our atomizer and atomization method. Our custom atomizer was constructed from machined aluminum following the designs of the TSI Model 3076 Constant Output Atomizer. We now describe the apparatus and operation of our atomizer in greater detail in the manuscript:

> The atomizer is constructed from machined aluminum and is based on the design of the TSI Model 3076 constant output atomizer (TSI, 2005). Briefly, filtered pressurized (30 psi) air is passed through a 0.01 inch critical orifice. Following the orifice, the air expands, causing seawater sample to be drawn up through inert polyethylene tubing and atomized by the jet of air. A polydisperse aerosol particle stream with a constant number and size distribution is created by the atomizer. The atomizer and tubing were sonicated with deionized water between samples to avoid cross-contamination. (Page 6 Lines 14-19)

3. Page 1, Line 31 - "Jet droplets aerosolized from the subsurface waters of highly productive regions may therefore be an unrealized source of effective INPs" should be removed since the authors have not probed jet droplets specifically.

We thank the reviewer for their suggestion. This reference to jet droplets has been removed. The sentence now reads:

Sea spray aerosol of composition similar to subsurface waters of highly productive regions may therefore be an unrealized source of effective INPs. (Page 1 Lines 31-32)

4. Page 3, Line 21 - The authors do an adequate job of introducing the process of natural sea spray formation in this paragraph. However, they have not introduced the mechanism by which they generate aerosols. This would be an ideal location to contrast the two aerosol formation approaches and the properties of the aerosols that result.

We now briefly state our method of aerosolization (referencing further details in the methodology section to follow).

These natural bubble bursting mechanisms contrast with laboratory methods of aerosolizing seawater. For instance, atomization – a technique employed in this study as well as previous ice nucleation studies (Ladino et al., 2016; Wilson et al., 2015) – does not result in the aerosolization of a microlayer and can result in aerosol particles with different compositions and size distributions to ambient SSA. Further discussion of the atomization technique employed here are described in the methodology section below. (Pages 3-4 Lines 36-2)

5. Page 3, Line 27 - The authors state the following: "SSA particles produced from jet drops are composed mainly of inorganic salts but may also contain whole or fragments of cells and soluble organic molecules in subsurface waters (Wilson et al., 2015; Wolf et al., 2019). Film burst particles can contain higher mass fractions of semi-soluble and insoluble organic molecules in the sea surface microlayer (Cochran et al., 2017)." While the authors are right to point out the current consensus that film droplets and jet droplets likely have distinct chemical characteristics I disagree with the use of solubility as a means of distinction. I would argue that there is consensus that it is the propensity of a molecule to go to the air-sea interface, that is surface-activity, that likely distinguishes which molecules are more likely to be present in the film droplets than the jet droplets and that solubility/=surface-activity when considering the plethora of organic compounds present in seawater. Two very similar compounds with equal surface-activity, both of which reduce interfacial free energy, can differ greatly in their behaviour because of a different degree of bulk solubility. Given this I would suggest the authors amend this statement.

We thank the reviewer for suggesting this reinterpretation. We have removed reference to solubility and instead discuss matter in terms of surface activity. These sentences now read:

The film burst and jet drop mechanisms can produce aerosols with distinctive chemical characteristics. This disparate composition results from differences in the surface activity – that is, the propensity of a molecule to go to the air-sea interface – of organic molecules. SSA particles produced from jet drops are composed mainly of inorganic salts but may also contain whole or fragments of cells and organic molecules with a low propensity to accumulate at the air-sea interface (Wilson et al., 2015; Wolf et al., 2019). Film burst particles can contain higher mass fractions of high surface-activity organic molecules in the sea surface microlayer (Cochran et al., 2017). (Page 3 Lines 28-34)

6. Page 3, Line 31 - The authors state that "The biogeochemistry of seawater can have a large impact on the composition of SSA". This is a generalisation that needs to be expanded upon with reference to the literature. The degree to which the composition of primary sea spray is affected by biological activity in the surface ocean is a long-standing question in the field. For example, recent field experiments where open ocean seawater were bubbled indicate that biological productivity has a minor influence on sea spray organic carbon content and composition (and its CCN properties for that matter) e.g. Bates et al., 2020; Quinn et al., 2014; Russell et al., 2010. Indeed, Beaupré et al. (2019) recently reported that highly aged DOM carbon could account for 19-40% of the organic carbon in artificially generated sea spray. In contrast,

Ceburnis et al. (2016) found that most organic enrichment in marine aerosol over the southern Indian Ocean was attributable to fresh POM. This dichotomy needs to be accurately represented in the introduction to the manuscript.

We thank the reviewer for suggesting several references and points to expand our discussion of seawater biogeochemistry. The paragraph following the referenced lines discusses how biogeochemistry can impact the diversity of phytoplankton (e.g. "Whereas upwelling zones and highly productive regions support larger phytoplankton species like diatoms and dinoflagellates, oligotrophic waters are characterized by different clades such as *Prochlorococcus* and *Synechococcus*).

The reviewer recommends a more nuanced discussion, however, of how biological activity in seawater can impact SSA composition. We have amended this paragraph by adding the following discussion:

> Research indicates a complex relationship between seawater biogeochemistry and the composition of SSA. Several recent field studies have indicated that rates of primary biological productivity have only a minor influence on the organic content of sea spray (Bates et al., 2020; Quinn et al., 2014; Russell et al., 2010). Other studies have indicated that aged organic matter, such as that metabolized by heterotrophic bacteria, are effectively transferred to the aerosol phase (Cochran et al., 2017; Wang et al., 2015). However, Beaupré et al., 2019 determined that up to 40% of the organic carbon in sea spray could be highly aged, and that the composition of SSA could be less strongly influenced by rates of primary biological productivity in the underlying seawater. Other studies have found that the organic enrichment of SSA is attributable to freshly produced fixed carbon, and that SSA carbon content is correlated with chlorophyl concentration (Ceburnis et al., 2016; O'Dowd et al., 2015). (Page 4 Lines 3-11)

7. Page 4, Line 6 - Since the authors state that " Measurements of INP concentration and activity from diverse marine regions are relatively rare" they should be able to provide an overview here. Given this some important recent literature is missing here (Creamean et al. 2019; McCluskey et al. 2018; Gong et al. 2020; Ickes et al. 2020).

We have removed the sentence stating that "measurements of INP concentration and activity from diverse marine regions are relatively rare." We have also referenced the suggested studies in an expanded discussion of findings from the recent literature. This paragraph discusses recent measurements in diverse marine regions, from tropical to high-latitude environments. The text now reads:

> Studies must investigate the cloud nucleation potential of SSA from diverse marine environments, including coastal, remote, high latitude, tropical, oligotrophic, and eutrophic ecosystems (Brooks and Thornton, 2018; Burrows et al., 2013). DeMott et al., 2015b investigated the ice nucleation activity of seawater from several remote locations, including the Caribbean, the oligotrophic Pacific, and the Bering Sea. Several other studies have focused on high latitude oceans, including the North Atlantic (Wilbourn et al., 2020; Wilson et al., 2015), Arctic (Ickes et al., 2020; Irish et al., 2017), and Southern Oceans (McCluskey et al., 2018b). Gong et al., 2020 investigated sources and concentrations of INPs in the seawater and atmosphere near the Cape Verde Islands, finding that SSA was only a minor source of INPs in this region. Creamean et al., 2019 demonstrated that biological productivity can infleunce INP concentrations in remote locations when organic material is transported along oceanic currents. These findings indicate the need to understand the sources and abundances of INPs in a diversity of marine environments. (Page 4 Lines 22-33)

8. Page 4, Line 14 - "This indicates that jet droplets in these regions may be an overlooked source of INPs" should be removed since the authors have not probed jet droplets specifically.

We thank the reviewer for articulating their reservations about our claim concerning jet droplet SSA. We have removed this sentence and similar ones (see comments below) from the text.

9. Page 4, Line 33 and Figure 2 - Sampling using a glass plate is a standard method in use since the early 70's. Given this it has been used in 100's if not 1000's of studies and there is absolutely no need to dedicate a figure in the main manuscript to it. As such, I suggest the authors either completely remove figure 2 or at the very least place it in the supplementary information.

We have moved this figure (Microlayer sampling with the plate) to the Supplemental Information section. We instead dedicate a figure in the main text to the aerosol size distribution generated from the atomizer (see comments below).

10. Page 4, Line 34 and Table 1 - The authors state that "rough seas precluded" collection of surface microlayer samples some distance away from the ship. Indeed table 1 shows that the average wind speed at the sampling locations was 15 m s$^{-1}$ and 13.5 m s$^{-1}$ in the Florida Straits and the Eastern Tropical North Pacific Ocean, respectively. These are very high wind speeds for sampling surface microlayer (experience tells me this was difficult!). Given this, I think some discussion on the potential impact of such rough seas on both the formation and persistence of the surface microlayer as well as the sampling is warranted here. For example, see the discussion in Rahlff et al. (2017),Sun et al. (2018), Engel et al (2018).

We thank the reviewer for raising this good point. It is indeed possible that high winds and rough seas at the time of sampling could have impacted the formation and persistence of the surface microlayer. We have added the following discussion to the text:

> Wind speed averaged 13.5 and 15 m s$^{-1}$ in the Florida Straits and ETNP, respectively, and at times exceeded 20 m s$^{-1}$ (Table 1). The resulting rough seas could possibly have impacted sea surface microlayer characteristics. For instance, Rahlff et al., 2017 determined that bacterial enrichment in the sea surface microlayer occurred only at winds speeds below approximately 5 m s$^{-1}$. Further studies have also determined a link between wind speed and the composition of the sea surface microlayer. Sun et al., 2018 determined that the abundance and size of macromolecular gels in a wave tank's microlayer decreased with winds above 8 m s$^{-1}$. Other studies have found an organic enrichment in the microlayer with wind speeds ranging from 10 to 13 m s$^{-1}$ (Sabbaghzadeh et al., 2017; Wurl et al., 2011), indicating that conditions were at times conducive to microlayer formation during our sampling. (Page 5 Lines 23-31)

11. Page 4, Line 37 - Although the authors have used a common approach to estimate the thickness of the sea surface microlayer they sampled, this number is highly uncertain and presenting it suggests higher confidence in it than is warranted. Given that this information is not at all critical to the later discussion I suggest the authors remove the following sentences "Based on the volume of seawater collected per dip and the surface area of the plate, the thickness of the organically-enriched layer adhering to the plate was on average 26µm. This falls within the range of previous findings (Irish et al., 2017)."

We have removed the indicated text from the revised manuscript.

12. Page 5, Line 5 - The following issue is certainly not limited to this study but should be mentioned here so that the authors and future readers of this manuscript interested in conducting similar experiments are aware. Given the high solubility of many of the surfactants enriched at the ocean surface a subsurface sample will rapidly form its own microlayer in a sample bottle or atomiser that may be very similar to a co-located micro-layer sample. For example, there is a significant body of literature presenting direct estimates of microlayer formation rates following disruption (e.g. Dragˇceviˇc and Pravdiˇc,1981, Kozaraca et al., 2005, Kuznetsova and Lee, 2001, Van-Vleet and Williams, 1983,Williams et al., 1986, Cunliffe et al., 2013) and

the current consensus is that they are rapid, typically<1min. This point further highlights the issue with the authors suggesting atomisation of their sub-surface samples can be equated with jet drop formation.

We again thank the reviewer for summarizing another issue regarding the atomization technique. Although thawed seawater samples were homogenized prior to atomization, we cannot preclude the possibility that organics in the subsurface sample partitioned at the surface during atomization. To address this point, we have added the following discussion to Section 2.3 – Seawater Aerosolization:

> We also note that the atomizer draws seawater from below the surface. Although our thawed seawater samples were homogenized with vigorous shaking prior to atomization organic partitioning at the surface occurs rapidly. Cunliffe et al., 2013 observed that the composition and bacterial makeup microlayer samples was reestablished only minutes after disruption. We therefore acknowledge the limits of our laboratory-generated data when it comes to drawing conclusions about ambient processes. (Page 5 Lines 32-37)

13. Page 5, Line 15 - The authors state that they use a "custom Collison-type atomizer" but do not provide any further information. Given the critical role this apparatus has to the study I would like to see either a reference to where it is described in detail or further details here. For instance, a schematic of the atomiser in the supplementary information would be much more useful than a schematic of glass plate sampling.

In response to this comment and comment 2 above, we have added the following description of our atomizer to the methods section.

> The atomizer is constructed from machined aluminum and is based on the design of the TSI Model 3076 constant output atomizer (TSI, 2005). Briefly, filtered pressurized (30 psi) air is passed through a 0.01 inch critical orifice. Following the orifice, the air expands, causing seawater sample to be drawn up through inert polyethylene tubing and atomized by the jet of air. A polydisperse aerosol particle stream with a constant number and size distribution is created by the atomizer. The atomizer and tubing were sonicated with deionized water between samples to avoid cross-contamination. (Page 6 Lines 14-19).

We have also included a reference to the instruction manual to the TSI Model 3076 constant output atomizer, which is identical to our atomizer.

14. Page 5, Line 21 - " Particles were size selected (mobility diameter = 200 nm)..." The authors state which size of particles were investigated in terms of the chemical composition and ice-nucleating ability but the reader has no sense of what the overall particle size distribution looked like given that none is presented. If the atomiser the authors used is anything like those I have encountered previously it will produce a narrow size distribution with relatively small particles. However, this is complete speculation until the authors present the size-distribution which they must do.

We have clarified several points in the text in response to this comment.

First, we have included a figure of the aerosol size distribution of the atomizer output (Figure 2). This indicates what the polydisperse size distribution looked like prior to size selection in our experiments. Also illustrated in Figure 2 is the diameter at which particles were selected prior to all ice nucleation and compositional measurements (200 nm). The reviewer is correct to point out the relatively narrow size distribution of the polydisperse aerosol generated by the atomizer.

We have also clarified that we used exclusively size-selected particles (200 nm diameter) in our experiments. This is now indicated in Figure 2 with a line at 200 nm, as well as several places in the text:

We investigated the composition of *200 nm* SSA generated from seawater samples using the Particle Analysis by Laser Mass Spectrometry (PALMS) instrument (Cziczo et al., 2006). (Page 7 Lines 22-23)

*Size-selected aerosol particles* were drawn into the nucleation chamber and nominally constrained to a flow centerline with particle free sheath air adjacent to the ice-covered walls. (Page 8 Lines 7-9)

15. Page 8, Line 14 - "Our compositional analysis demonstrates that the ocean biogeochemistry impacts the composition of SSA". Given the actual experiments conducted by the authors the language used in this sentence is far too strong. The analysis conducted by the authors demonstrates that aerosols generated by an atomiser from seawater with very different biogeochemical states have differing composition.

We have reworded this sentence to more accurately describe our findings:

Our compositional analysis demonstrates variability in the composition of our laboratory-generated aerosol particles. (Page 10 Line 14)

16. Page 9, Line 15 - "This indicates that both jet drop particles originating from subsurface water and smaller film burst particles originating from the sea surface microlayer in productive marine environments can be effective depositional INPs. These organically-enriched jet droplets can constitute a large fraction of submicrometer SSA (Wang et al.,2017)" should be removed since the authors have not probed jet droplets specifically.

We have removed these sentences from the revised manuscript.

17. Page 10, Line 12 - "Atomizing seawater creates SSA with more uniform composition than natural seawater aerosolization processes, as it does not mimic the film burst and jet drop aerosolization processes that create organically enriched and depleted SSA, respectively." Here the authors have nicely summarized the major issue with the manuscript in its current form. This discussion belongs much earlier in the manuscript alongside the introduction of the process of film and jet droplet production in natural bubble bursting (see my comments above). Also, I would like to see a reference for the statement "Atomizing seawater creates SSA with more uniform composition than natural seawater aerosolization processes...". Do the authors have evidence for this or is it simply speculation? It is critical when it comes to the next point.

The reviewer raises two points that we have revised our manuscript to address. First, the reviewer requests that we discuss the limitations of the atomization aerosolization technique earlier in the manuscript. We have now done so by expanding the discussion in Section 2.3 – Seawater Aerosolization. We refer to our response in comments 1, 4, and 12 above. In particular, we now explicitly discuss the limitations of the atomization technique when it comes to mimicking the composition of ambient SSA.

Second, the reviewer requests clarity on our claim that "atomizing seawater creates SSA with more uniform composition than natural seawater aerosolization processes." We have revised this to say that "Atomizing seawater creates aerosol particles less enriched in organics than natural seawater aerosolization processes…," and we now cite Gaston et al. 2011, which compared the effects of aerosolization technique (atomization versus bubbling) on resulting SSA composition. Specifically, atomizing seawater was found by Gaston et al. to produce fewer organically-enriched particles than bubbling seawater. We have expanded on this in the text by adding the following discussion:

Atomizing seawater creates aerosol particles less enriched in organics than natural seawater aerosolization processes, as it does not mimic the film burst and jet drop aerosolization processes that create organically-enriched and depleted particles, respectively. For instance, Gaston et al., 2011 observed that atomizing seawater produces over 27% fewer organically-enriched particles compared to bubbling. The majority of 200 nm particles in ambient SSA arise from the film-burst production process (Pruppacher and Klett, 1980). Wang et al., 2017 determined that film-burst particles constitute at least 57% of submicron SSA, with the remainder resulting from jet droplets. Our atomized SWPs are likely less organically-enriched than ambient SSA. (Page 12 Lines 10-16)

18. Page 10, Line 16 - "Our derived ns values may therefore be lower estimates for immersion mode INP activity." Following on from my previous point, given that the authors provide no evidence suggesting that atomized seawater has a more "uniform composition than natural seawater aerosolisation processes" this sentence is idle speculation and should be removed. It would be equally unjustified for me to say that the narrow size distribution with small particles that are likely more enriched in organic material compared to larger particles sizes will bias estimated ice nucleation site densities to higher values compared to natural aerosol. Without further information we cannot say either way.

We thank the reviewer for bringing this discrepancy to our attention. As highlighted above, we now discuss in our text the ways in which the atomization process differs from ambient SSA formation mechanisms.

We state that while the atomization process produces fewer organically-enriched aerosol than natural bubble bursting mechanisms, it is an energetic process that can rupture cells and possibly lead to an artificial increase in INP-active macromolecules from within cells. Since we cannot definitely say whether the atomization technique will increase or decrease ns values, we have revised our text accordingly:

> At this time, it is unknown whether the atomization technique results in a greater or lesser ns density compared to natural SSA formation mechanisms. (Page 12 Lines 18-20)

19. Page 10, Line 34 - "Both film burst and jet droplet particles generated from microlayer and subsurface waters in productive regions such as the ETNP are likely to be sources of effective INPs. In less productive regions, film burst particles may be the dominant source of marine INPs." This statement may well be true but the authors have not generated data that would allow them to test this so both these sentences must be removed.

We have removed this sentence from the revised manuscript.

20. Page 11, Line 12 - "The subsurface is aerosolized through bubble bursting mechanism, which create jet droplets (Pruppacher and Klett 1980, Wilson et al. 2015, Wang et al. 2017). This implies that jet droplet aerosols generated in coastal upwelling regions or during spring phytoplankton blooms can be a source of INPs." Again, the authors have not generated data that would allow them to test this so both these sentences must be removed.

We have removed this sentence from the revised manuscript.

21. Page 11, Line 34 - "However, our results demonstrate that larger jet drop particles originating from highly productive subsurface waters may be a source of effective INPs as well." Again, the authors have not generated data that would allow them to test this so both these sentences must be removed.

We thank the reviewer for highlighting the several instances where we have erroneously referred to atomized subsurface water as "jet droplets." Here and in other instances, we have removed indicated text from the manuscript.

**Reviewer 2**

The study examines laboratory generated particles from samples of seawater and the surface microlayer from two different locations wrt. their ice nucleation ability. It is an interesting study, showing that oceanic productivity and ice nucleation ability of the related particles are somewhat connected. It is suggested that jet droplets occurring during sea spray production might play a larger role for atmospheric INP, which, however, is not really examined in the study itself, as all examined particles are generated artificially and the sea spray particle generation mechanism was not examined at all.

We thank the reviewer for their accurate summary of the manuscript's findings. Below, the reviewer outlines several concerns regarding the clarity of the manuscript as well as our interpretation of the data. We highlight several substantial revisions which we believe address the reviewer's concerns.

2. While the topic of the paper is interesting, writing needs to be improved in a number of locations. Particularly the introduction needs to be improved a lot. It does not really focus on marine INP (which it should have), but instead is a broad collection of information given in detail which would not need to be so detailed (such as different types of INP or mixed phase and cirrus clouds). This contrasts with the fact that publications dealing with topics related to the content of this manuscript are missing. The focus of this introduction does not fit to the scope of the manuscript. Comments on the "Introduction"-section are therefore given separately below.

   As detailed below in our response to comments $4-13$, we have made several revisions and additions to the introduction of our manuscript which we believe now accurately summarizes the (most recent) literature on ice nucleation of marine aerosol. We do retain, however, background information on ice nucleation in general. This includes a brief description of the modes of heterogeneous ice nucleation, as well as a description of ice nucleation in mixed phase and cirrus clouds. We hold that this information is critical to motivate our and all research into ice nucleation.

3. Another more general concern is the use of the word SSA (sea spray aerosol) for the particles examined here. SSA is a specific aerosol generated by wave activity and bubble bursting - and then an aerosol always includes particles as well as the gas-phase around them. Strictly speaking, the study examines particles generated from sea water samples. To avoid confusion, I would recommend using SWP (sea water particles) or such. Also, it needs to be check throughout the text if it is referred to particles or really all of the aerosol. Generally, the use of "P" (particle) instead of "A"(aerosol) will be better. Once these issues, together with the other more detailed ones below will have been addressed, the manuscript can be considered for publication in ACP. But a thorough revision of the manuscript is needed at first.

   We thank the reviewer for their suggestion to differentiate between natural, ambient sea spray aerosol (SSA) and the aerosol we generated in our laboratory setting. The reviewer also indicates that they recommend distinguishing between "aerosols" and "particles." We note that "whereas an aerosol is technically defined as a suspension of fine solid or liquid particles in a gas, common usage refers to the aerosol as the particulate component only (Seinfeld and Pandis, 1998)." We nonetheless adopt the reviewer's recommendation to substitute SSA with sea water particles (SWPs) when referring to our laboratory-generated aerosol:

   > In this study, we identify a link between primary productivity in marine environments and the INP activity of particles generated from seawater in a laboratory setting, which we refer to as sea water particles (SWPs). (Page 4 Lines 35-36)

This correction has been made throughout the text. We retain the acronym SSA, however, when referring to ambient aerosol.

**Introduction:**

4. p2, lines 8-11: These two sentences (starting with "Ice formation" and ending with "important") don't make sense together. Ice-formation (meaning the mechanism of ice nucleation) is NOT the Wegener-Bergeron-Findeisen effect. The latter concerns growth of ice crystals even in regions that have relative humidities < 100% wrt. liquid water. These sentences need to be completely reformulated.

We thank the reviewer for drawing our attention to this ambiguity. We have reformulated these sentences to clearly distinguish between the initiation of ice nucleation and the Bergeron-Findeisen process. The passage now reads:

> Ice formation in mixed-phase clouds is important for initiating precipitation. Ice crystals grow by scavenging water vapor from liquid droplets through the Wegener-Bergeron-Findeisen process, increasing the settling velocities of ice particles (Pruppacher and Klett, 1980). This effect decreases cloud lifetime and is responsible for over 70% of precipitation globally (Lau and Wu, 2003). (Page 2 Lines 8-11)

5. p2, line 17: Citing Whale et al. (2018) for this is awkward as this is textbook knowledge.

The indicated citation (Whale, 2018) refers to a textbook chapter. We now cite the entire textbook (Andronache, 2018) in addition to Pruppacher and Klett to guide readers to these broad overviews on ice nucleation in mixed-phase and cirrus clouds.

6. p2, lines 17-30: The different heterogeneous ice nucleation mechanisms are not clearly described, and their importance is not mentioned (immersion freezing is thought to be the most important for mixed phase clouds, for cirrus this is not clear yet). Instead, remarks are made on comparably unimportant effects. This needs to be rewritten.

We have added more clarity to our description of the different modes of heterogeneous ice nucleation. Specifically, we have streamlined the paragraphs describing heterogeneous nucleation by focusing exclusively on the two mechanisms considered here: depositional and immersion ice nucleation. As per this comment and comment 7 below, we remove our discussion on pore condensation freezing and deliquescence freezing. We have also discussed the importance of homogeneous, depositional, and immersion mode ice nucleation in terms of different cloud systems. The text now reads:

> Several pathways of heterogeneous ice formation exist. Depositional ice nucleation occurs above ice saturation but below liquid water saturation when ice deposits directly onto the solid surface of an INP. Depositional ice nucleation and homogeneous freezing are the two predominant pathways for cirrus cloud formation (Barahona et al., 2010; Cziczo et al., 2013; Kärcher, 2017; Lohmann et al., 2004). Immersion freezing can occur above liquid water saturation when an INP first activates as a cloud condensation nucleus. This process is important for ice formation in mixed-phase clouds (Murray et al., 2012; Pruppacher and Klett, 1980). (Page 2 Lines 18-23).

7. p2, line 20: "pore condensation and freezing" was first suggested and examined in Marcolli (2014), so it would be fair to cite that publication here. Or to skip that mentioning completely, as this is not what you are looking at.

By streamlining our discussion of the mechanisms of heterogeneous ice nucleation, we have adopted the reviewer's advice and removed reference to pore condensation freezing.

8. p2, lines 33-37: There is no need to go into so much detail for types of INP which are certainly NOT emitted by the ocean. It is also somewhat unclear which citation here is given for which type of INP. Also, there are good reviews which you could cite instead, two of which you already used above (Hoose & Möhler 2012; Murray et al., 2012), but also a much older one (Szyrmer and Zawadzki, 1997) and a newer one (Kanji et al.,2017) - it would be better to cite reviews here instead of your selection, which often does not include the oldest / newest / most cited publication for the separate INP types, anyway, and which is too detailed, given your focus on marine INP.

We thank the reviewer for their suggestion to reduce our discussion of terrestrially-sourced INPs. To clarify our text, we have removed reference to specific sources of terrestrial INPs. We instead briefly mention that INPs come from terrestrial sources, citing several overview papers suggested by the reviewer above. The text now reads:

> Despite their climatic importance, the sources and characteristics of atmospherically relevant INPs remain uncertain. Laboratory and field studies have identified several terrestrially-sourced INPs (Hoose and Möhler, 2012; Kanji et al., 2017). Characterizing marine sources of INPs also remains an active area of research (Brooks and Thornton, 2018; Kanji et al., 2017). (Page 2 Lines 28-31)

9. p3, line 1: You miss all the new work on that, which should not have happened, given that this is the topic you are focusing on in here. It's weird that here now you cite review papers, on the topic you want to look at in depth. Just a selection: Burrows et al.(2013), Creamean et al. (2019), Gong et al. (2020), Ladino et al. (2019), McCluskeyet al. (2018a,b).

We thank the reviewer for suggesting articles to discuss and cite in our introduction. We discuss many of these articles in several contexts throughout the manuscript. In particular, we have revised the introduction to reflect the findings of these recent studies. We now discuss the findings of several of these studies in the context of the introduction's literature review:

> Several studies have sought to clarify the importance of marine versus terrestrial INP sources. Ladino et al., 2019 reported that biological particles of possible marine origin were an important source of warm-temperature immersion INPs at a tropical site on the Gulf of Mexico. In contrast, Gong et al., 2020 investigated sources and concentrations of INPs in the seawater and atmosphere near the Cape Verde Islands, finding that SSA was only a minor source of INPs in this region. McCluskey et al., 2018b found that enhanced primary productivity does not necessarily enhance the concentration of INPs in the marine boundary layer. Other studies have sought to parameterize and model the ice nucleation activity of marine INPs. A recent parameterization by McCluskey et al., 2018a demonstrates that nascent SSA exhibits $1/1000^{th}$ of the ice nucleating active sites per unit surface area compared to mineral dust. Global model outputs indicate that SSA may nonetheless be an important source of INPs in remote regions away from terrestrial aerosol inputs (Burrows et al., 2013; Vergara-Temprado et al., 2016). (Page 3 Lines 13-22)

10. p3, line 8: Also the link from the ocean to the atmosphere is important for your claim, and that is understudies, too! Particularly three recent publications (already included above) might be important in this respect, as they are dealing with marine INP (which necessarily includes sea spray): McCluskey et al. (2018a,b) find low INP concentrations in remote marine regions (Southern Ocean and North East Atlantic), Gong et al.(2020) find that marine INP contribute only a very small fraction of atmospheric INP in Cape Verde. As these publications are directly linked to your topic they should be discussed in your work. Also the above mentioned publication by Creamean et al. (2019) might be of interest in that respect.

The reviewer raises a good suggestion to highlight complex link between the ocean and the atmosphere. We have added a paragraph of text to the introduction highlighting the relationship between seawater biogeochemistry and the composition of SSA:

> Research indicates a complex relationship between seawater biogeochemistry and the composition of SSA. Several recent field studies have indicated that rates of primary biological productivity have only a minor influence on the organic content of sea spray (Bates et al., 2020; Quinn et al., 2014; Russell et al., 2010). Other studies have indicated that aged organic matter, such as that metabolized by heterotrophic bacteria, are effectively transferred to the aerosol phase (Cochran et al., 2017; Wang et al., 2015). However, Beaupré et al., 2019 determined that up to 40% of the organic carbon in sea spray could be highly aged, and that the composition of SSA could be less strongly influenced by rates of primary biological productivity in the underlying seawater. Other studies have found that the organic enrichment of SSA is attributable to freshly produced fixed carbon, and that SSA carbon content is correlated with chlorophyl concentration (Ceburnis et al., 2016; O'Dowd et al., 2015). Aside from organic mass fraction, seawater biogeochemistry can also affect the speciation of organic molecules in SSA. Regions of high primary productivity, such as upwelling environments or springtime phytoplankton blooms, exhibit different planktonic species than regions with low primary productivity (Righetti et al., 2019). (Page 4 Lines 3-14)

In addition, we have added discussion on the geographic dependency of the air-sea interface the reviewer highlights:

> Studies must investigate the cloud nucleation potential of SSA from diverse marine environments, including coastal, remote, high latitude, tropical, oligotrophic, and eutrophic ecosystems (Brooks and Thornton, 2018; Burrows et al., 2013). DeMott et al., 2016 investigated the ice nucleation activity of seawater from several remote locations, including the Caribbean, the oligotrophic Pacific, and the Bering Sea. Several other studies have focused on high latitude oceans, including the North Atlantic (Wilbourn et al., 2020; Wilson et al., 2015), Arctic (Ickes et al., 2020; Irish et al., 2017), and Southern Oceans (McCluskey et al., 2018b). Gong et al., 2020 found that INPs were both enriched and depleted in the sea surface microlayer relative to subsurface water near the Cape Verde Islands, indicating the effects of both transient biological activity as well as physical parameters such as ocean mixing. Creamean et al., 2019 demonstrated that biological productivity can influence INP concentrations in remote locations when organic material is transported along oceanic currents. These findings indicate the need to understand the sources and abundances of INPs in a diversity of marine environments. (Page 4 Lines 22-33)

We thank the reviewer for their suggested citations, which have been reflected in the added text above.

11. p3, line 9: "DeMott et al. (2015)": I guess you mean the one that is given as "(2015b)" in your references? But that actually is "(2016)", anyway. (The preprint came out shortly before new years in 2015, but the final printing date was in 2016).

We thank the reviewer for pointing out this error! We have corrected the reference to DeMott et al. 2016 in both the text and the references.

12. p3, lines 14-16: Is the content of this sentence related to Wilson et al. (2015) (which you cite in the beginning of the sentence before) or to Knopf et al. (2011) (which you cite at the end of the next sentence)? Clarify!

The numbers cited here refer to Wilson et al. 2015. We have appended this citation to the end of this sentence.

 p3, line 21: "SSA encompasses a range of particle chemistries": SSA does not "encompass" particle chemistries. The chemistry goes on in the SML or sea water. Reformulate! Or if you want to point to the next sentence, then it would be "The formation of SSA encompasses a range of physical processes that affect / are affected ... ."

We have adopted the reviewer's recommended construction, and reformulated this sentence to read:

> The formation of SSA encompasses a range of physical processes that affect ice nucleation ability. (Page 3 Line 23)

General comments:

14. p5, line 15: Give more information on the working principle of the atomizer. Your readers need to know how the particles were generated. The generation process is a big part of atmospheric SSA, in terms of particle sizes, particle concentrations and particle composition, and this is not easily reproduced with just generating particles from sea water or SML samples. You can check for some information on this issue in the introduction of Fuentes et al. (2010). "Real-world-line" SSA likely is best obtained by using wave channels (Prather et al., 2013). The generation technique you use is rather just a means to generate particles, but if they are similar to atmospheric particles generated from sea spray is a separate issue.

We have substantially revised the manuscript in response to this comment and several comments by Reviewer 1. We summarize the changes made here.

First, we provide more details on the atomizer in Section 2.3 – Seawater Aerosolization:

> The atomizer is constructed from machined aluminum and is based on the design of the TSI Model 3076 constant output atomizer (TSI, 2005). Briefly, filtered pressurized (30 psi) air is passed through a 0.01 inch critical orifice. Following the orifice, the air expands, causing seawater sample to be drawn up through inert polyethylene tubing and atomized by the jet of air. A polydisperse aerosol particle stream with a constant number and size distribution is created by the atomizer (Figure 2). The atomizer and tubing were sonicated with deionized water between samples to avoid cross-contamination. (Page 6 Lines 14-19)

N.B. that we now reference the user manual for the TSI Model 3076 constant output atomizer, on which our in-house atomizer was based.

We also acknowledge in several places that the atomizer likely produces aerosol particles that differ from ambient sea spray aerosol. For instance, in the introduction, we state:

> These natural bubble bursting mechanisms contrast with laboratory methods of aerosolizing seawater. For instance, atomization – a technique employed in this study as well as previous ice nucleation studies (Ladino et al., 2016; Wilson et al., 2015) – does not result in the aerosolization of a microlayer and can result in aerosol particles with different compositions and size distributions to ambient SSA. The atomization technique employed in this study is further described in the methodology section below. (Pages 3-4 Lines 36-2)

We further discuss the limitations of the atomization technique – referencing several of the articles the reviewer recommends, in our revised Section 2.3:

We note that the atomization technique – although used in prior studies investigating the ice nucleation of sea spray aerosol (Ladino et al., 2016; Wilson et al., 2015) – has several limitations. Specifically, atomization produces aerosol with different physical and chemical characteristics than ambient SSA (Collins et al., 2014). First, atomization results in a different aerosol size distribution due to the lack of bubble bursting mechanisms (Fuentes et al., 2010). The impact of this artefact can be limited by size-selecting a narrow diameter range from the resulting polydisperse aerosol stream prior to INP analyses. However, atomization also results produces aerosols of a different composition than ambient SSA (Gaston et al., 2011). Natural bubble-bursting mechanisms result in aerosol with size-dependent composition (Collins et al., 2014; O'Dowd et al., 2004; Prather et al., 2013). It is unlikely that atomization can replicate the composition of natural SSA. Further, atomization is an energetic process that may result in a higher rate of cell lysis than expected from natural processes, such as apoptosis, viral infection, or predator grazing (Agustí and Duarte, 2013; Kirchman, 1999). This may artificially increase the organic content of our laboratory-generated aerosol and increase the occurrence of ice nucleating macromolecules in particles (Ickes et al., 2020; Knopf et al., 2011). We also note that the atomizer draws seawater from below the surface. Although our thawed seawater samples were homogenized with vigorous shaking prior to atomization organic partitioning at the surface occurs rapidly. Cunliffe et al., 2013 observed that the composition and bacterial makeup microlayer samples was reestablished only minutes after disruption. We therefore acknowledge the limits of our laboratory-generated data when it comes to drawing conclusions about ambient processes. (Page 6 Lines 21-37)

We believe these revisions provide adequate details on our atomizer and sufficiently acknowledge its limitations.

15. p5, line 21: Choosing a mobility diameter of 200nm implies that you assume that marine INP are all separately floating (likely biogenic) macromolecules. Mention that explicitly, and elaborate on that - that is one thing that could also be discussed already in the introduction. You need to justify why you can assume that this choice will not cause you to lose the majority of all INP (see Wilson et al. (2015) and Irish et al. (2017), which you already cite).

We thank the reviewer for raising this point. We have expanded our discussion on our choice to size-select 200 nm particles in Section 2.3. We have included the following text that highlights the majority of marine-derived INPs can still be found in 200 nm particles:

The particle diameter was chosen to align with previous experiments' methods (DeMott et al., 2016; Wilson et al., 2015), yet we acknowledge INP activity varies with SSA diameter (DeMott et al., 2016; Si et al., 2018). This choice reflects previous findings that marine INPs are likely macromolecular organic clusters smaller than 200 nm. For instance, Irish et al., 2017 quantified INP size in Arctic seawater samples, identifying that the majority of immersion-mode INPs in seawater were between 20 and 200 nm. Wolf et al., 2019 further demonstrated that a variety of marine-derived molecules smaller than 200 nm were INP active in the depositional ice nucleation mode. A likely source of these molecular INPs are phytoplankton exudates (Ickes et al., 2020; Knopf et al., 2011; Wilson et al., 2015). (Page 7 Lines 12-19)

16. p5, lines 22-24: Doubly and triply changed 200nm particles should still be smaller than 500nm, so the choice of your cut-off might not have been optimal. You can still argue that this will remove the more highly charged particles (which, however, do not occur in such high amounts). Please check this, and also correct the text accordingly, so that others will not copy this (wrong) approach in the future. An estimation on the influence of multiply charged particles on the final results is needed.

To address this point, we have added a new figure to the text demonstrating the size distribution of the polydisperse aerosol prior to size selection in the DMA. The relatively narrow size distribution results in many more 200 nm particles than particles of diameters corresponding to multiply charged particles (Figure 2). We estimate that the contribution of doubly charged particles to the total number concentration of sampled particles is less than 3%. The text has been updated to reflect this:

> A 500 nm size cutoff impactor was used upstream of the DMA to prevent large multiply-charged particles from entering the sampled aerosol stream. Nonetheless, doubly charged particles may have been sampled. Figure 2 illustrates that the concentration of 400 nm particles is approximately 11% that of the 200 nm particles we size selected. Given that the ratio of doubly to singly charged particles predicted by a Fuchs charging model applied to a DMA neutralizer is below 0.3 (Mamakos, 2016), we estimate that multiply charged particles constitute only less than 3% of the total particles sampled. (Page 7 Lines 7-12)

17. p7, line 6: This was mentioned above, but again (and valid for the whole text): The use of "SSA" for artificially generated aerosol is a bit misleading.

We again thank the reviewer for their recommendation to use Sea Water Particles (SWPs) rather than Sea Spray Aerosol (SSA) when referring to our laboratory-generated particles. We have corrected the text throughout to reflect this recommendation.

18. p7, lines 19-26: Check this whole paragraph, together with Fig. 4. There are inconsistencies! I think that the following could remove these issues, but please check carefully for yourself: (a) line 21 -> change 4d to 4a! (b) line 25 -> change 4a to 4d! (c) "This suggests that subsurface waters in the ETNP supported more primary production." – I assume you mean compared to the Florida Straits? Then please say so explicitly!

We have corrected the references to the panels in Figure 4. We also thank the reviewer for suggesting we explicitly say that we compare the ETNP to the Florida Straits. We have done so in the revised text:

> This suggests that elevated primary productivity in the ETNP sustained organic carbon content in the subsurface waters more so than in the Florida Straits. (Page 9 Lines 15-16)

19. p7, lines 28-29: Comparing averages here might seem the wrong choice, as the higher values for ETNP come from two outliers (at least to some extent). However, also if you gave the median values, your statement would still be correct (as far as I can see).Therefore, it might be worth adding the median values as well.

We have now included both the average and the median values of subsurface organic carbon signal. These coupled sentences now read:

> The average organic carbon signal for the ETNP subsurface samples was $1.11 \pm 0.62$ ($1\sigma$ variability), whereas the average organic carbon signal for the Florida Straits subsurface samples was $0.41 \pm 0.20$… Likewise, median values are 0.76 and 0.29 for the ETNP and Florida Straits subsurface samples, respectively. (Page 9 Lines 22-25)

20. p8, lines 5-6: "PALMS detecting ionization of more soluble organic nitrogen species in addition to amino acids." - And you assume that one would be present in higher concentration in the subsurface, the other one more in the microlayer water? Clarify! Also: Which type of detection was used by Zäncker (previous sentence)? And are you aware of the following publications: Kuznetsova & Lee (2002), Kuznetsova et al.(2004), Reinthaler et al. (2008) and Engel & Galgani (2016), who all found

amino acids enriched in the surface microlayer in different oceanic regions. This may support your hypothesis, that the ionization efficiency of different compounds in PALMS might influence your results, which, however, then would have an influence on the interpretation of your data. Enrichment in the surface microlayer is not my main expertise, so take this merely as a suggestion for something you could look into.

Here the reviewer raises several points, which we discuss in turn below.

First, we have clarified that different nitrogenous molecules may partition in different places on account of their solubility or surface activity:

> PALMS' detection of more soluble organic nitrogen species in subsurface waters in addition to certain amino acids that partition in the microlayer could have led us to observe parity of organic nitrogen in both microlayer and subsurface samples. (Page 10 Lines 2-4)

We thank the reviewer for indicating several additional studies which also found an enrichment of amino acids in the microlayer. We have included these references in the text:

> Several previous studies have found that amino acids are enriched in microlayer samples relative to subsurface from both coastal and remote waters (Engel and Galgani, 2016; Kuznetsova et al., 2004; Kuznetsova and Lee, 2002; Reinthaler et al., 2008; Zäncker et al., 2017). Page 9-10 Lines 36-1)

Finally, we allude to the effects of matrix effects and PALMS ionization efficiencies in both Section 2.4 – Chemical Characterization – and Section 3.1 – Seawater Chemistry:

> Particle ionization with the UV excimer is not quantitative (Cziczo et al., 2006; Murphy et al., 1998). However, the average relative intensity of organic signal in a sample's mass spectra can qualitatively indicate which seawater samples are organically-enriched (Wolf et al., 2019). (Page 7 Lines 33-35)

> Several factors, such as matrix effects and variable ionization efficiencies of different molecules, can affect the observed signal in PALMS mass spectra (Cziczo et al., 2006; Murphy, 2007; Murphy et al., 2006; Zawadowicz et al., 2017). (Page 10 Lines 4-6)

21. p8, line 15: "similar and elevated organic contents" - well, this is not true for the micro-layer, which has quite similar values for ETNP and Florida Straits. Only the subsurface value is higher for ETNP than for Florida Straits.

We have removed "elevated." The sentence now reads:

> Aerosols generated from both the subsurface and microlayer samples from the highly productive ETNP contained similar organic contents. (Page 10 Lines 15-16)

22. p8, lines 17-18: This last sentence of this paragraph might not be true for INP. As we don't know what it is that makes the INP, this conclusion cannot be drawn! Remove!

We have removed the indicated sentence from the revised manuscript.

23. p9, lines 5-7: Zeppenfeld et al. (2019) finds a connection between glucose and INP concentrations in sea water samples, for real world measurements on Arctic samples. Please add.

We have added Zeppenfeld et al. 2019 to the references here, as well as indicated that carbohydrates (like glucose) may impact ice nucleation activity.

24. p9, lines 17-18: I am not sure if all readers will be familiar with the study by Wang et al. (2017) which you relate to here. As this is very important for this paragraph and also for conclusions you make later, explicitly describe somewhere in your text (maybe in this paragraph here), that Wang et al. (2017) claims that jet droplets can make up a substantial fraction of all sea spray generated particles. It would be good to also be more specific instead of just writing "a large fraction".

We have now expanded on the findings of Wang et al., specifically mentioning the percentage of SSA resulting from jet droplets versus film-burst processes. The text reads:

> Wang et al., 2017 determined that film-burst particles constitute at least 57% of submicron SSA, with the remainder resulting from jet droplets. (Page 12 Lines 14-16)

25. p10, equation 1: This is an approximation which only works for really small fi (∼up to 0.1, as they are typically measured for example in the AIDA cloud chamber – but I guess you might have had higher fractions in SPIN). Otherwise, the full exponential has to be used!

We thank the reviewer for suggesting this clarification. We presume the exponential the reviewer mentions refers to the equations of DeMott et al., 1994. We first note that these equations were developed to account for multiple INPs in aliquot droplets on cold stage apparatus. Our sampled volume in the online technique SPIN (200 nm particles) are orders of magnitude smaller than the volumes typically sampled with these apparatus (∼50 μL aliquots), which greatly reduces the probability that we have sampled multiple INPs in one droplet.

In calculating INAS densities from our immersion freezing results, we rely on fractional activation values that result from aerosol spreading outside of the SPIN lamina (See e.g. Garimella et al., 2017 and our discussion in Section 2.5 – Ice Nucleation Measurement:

> Turbulent mixing near the aerosol inlet causes particles to spread outside of the aerosol lamina. This exposes particles to a wider temperature range and lower $S_{ice}$ than that of the lamina centerline (Garimella et al., 2017). Particles outside of the lamina are therefore less likely to activate as INPs. To account for this artefact, a correction factor is normally applied to measured INP and fractional activation data (DeMott et al., 2015a; Garimella et al., 2017; Wolf et al., 2019). We apply the methods and correction factors detailed in Garimella et al. (2017) and Wolf et al. (2019) to immersion and deposition nucleation data presented in this study. (Page 8 Lines 16-21)

In keeping with standard practices, higher fractional activation values were achieved by applying these correction factors to our observed immersion freezing data. We have clarified in the text that Equation 1 is an approximation applicable to small fractional activation values. Our addition reads:

> Equation 1 is an approximation applicable to small fractional activations. We use fractional activation values corrected for aerosol spreading outside the central lamina (DeMott et al., 2015; Garimella et al., 2017), as detailed in Section 2.5 above. (Page 12 Lines 6-9)

26. p10, lines 32-33: Which surface area did DeMott use? (Hint: it is different from how you did it.) You should say explicitly in your study that the total marine aerosol includes a lot of other particles and that the n_s value you give here cannot simply be used for upscaling!

We thank the reviewer for suggesting this important clarification. We have included in the text a statement that our INAS ($n_s$) values cannot be extrapolated due to the complexity of natural and ambient SSA:

> However, we caution that our derived $n_s$ values should not be used to extrapolate $n_s$ for ambient marine aerosol, as SSA may differ in composition to our SWPs and total marine aerosol includes many particle sources not considered here, such as secondary aerosol sources (Facchini et al., 2008; Fu et al., 2013; O'Dowd and de Leeuw, 2007). (Pages 12-13 Lines 37-2)

27. p11, lines 4-5: Here you can see very clearly that my earlier suggestion to use another name for the particles you look at, instead of SSA, makes sense. An aerosol cannot be an INP - a particle can be an INP. So in any case, "aerosols" in line 5 should become "particles". Please check the whole text to see if you really refer to aerosol only where you mean the combination of particles with the surrounding air!

We have amended the text to refer to our laboratory-generated particles as "Sea Water Particles" (SWPs).

28. p11, lines 22-24: And yes, not only modeling studies show that, but also some of those publications using real world data which I suggested above. This should be added here.

We thank the reviewer for their suggestion to cite recent field studies in our discussion here. We have included a sentence summarizing the findings of several of the studies the reviewer indicated in their previous comments:

> Several recent field studies have also indicated the potential importance of marine sources of INPs in both remote and coastal atmospheres (Creamean et al., 2019; Ladino et al., 2019; McCluskey et al., 2018a). (Page 13 Lines 24-26)

29. p11, lines 29-30: "...(McCluskey et al., 2018). This demonstrates that even highly productive marine environments are not always effective sources of INPs." This is misleading - it depends on the interpretation of "effective". If no (typically strong) land sources are nearby, then the marine environment is the only remaining source, and therefore it could be "effective". Reformulate!

We have reformulated this sentence in a way that no longer compares the effectiveness of different INP sources. It now reads:

> This demonstrates that INP concentrations are not uniformly elevated in highly productive marine environments. (Page 13 Lines 31-32)

**Specific comments:**

30. p2, line 14: last word "from" -> should be "form"
31. p3, line 24: "eject" -> "ejects"
32. p3, line28: Shouldn't "in" be "included in" or "from"?
33. p7, line 8: Following "signals", add "with PALMS".
34. p8, line 2: "2017" -> "(2017)"
35. p8, line 5: "(Figures 4b and 4e)" should only be "(Figure 4e"), or you should also mention ETNP in this sentence.
36. p9, line 9: "2015" ->"(2015)"
37. p11, line 18: "that that" - remove one of them.

We thank the reviewer for their multiple suggestions here to improve the clarity or grammar of the manuscript. Each of these minor changes have been enacted in the revised manuscript.

---

## Author Response (AR3)

Dear Editor,

We again thank the reviewers for their additional comments and suggestions regarding our manuscript. We are pleased to read in their reports that our previous revisions have satisfied their concerns and improved the manuscript. We also thank you, the Editor, for your suggestions and efforts to see our manuscript through the peer-review process.

Our point-by-point response to reviewers' and Editor's comments is included below. For clarity, comments are in green text and our responses are in black. Our line numbers reference the updated document, not the original.

**Editor's Comments**

Many thanks for your thorough revisions. The manuscript has improved a lot, which is also acknowledged by the two reviewers. Both reviewers have a few more minor remarks which I would like you to address before final acceptance.

We again thank the Editor for his suggestions to improve our manuscript before final acceptance.

I agree with reviewer #2 that it would be wise to adapt the title given the specific particle production method applied here.

We have adapted the title to remove reference to sea spray aerosol, as requested by reviewer 2. Our manuscript is now entitled "A Link between the Ice Nucleation Activity and the Biogeochemistry of Seawater."

Figure 2 (the size distribution of the atomizer) is not really needed in the main manuscript and is probably better suited for the SI.

We apologize for misunderstanding the original request of the reviewer. We agree that the figure depicting the size distribution of the atomizer is best suited for placement in the Supplemental Information section. We have moved it there and renumbered the figures accordingly.

Please also clarify which fit method was used in Fig 4 (I assume it was an orthogonal one?).

We thank the Editor for drawing our attention to this omission! We now indicate the method of linear regression in both the main text and the figure caption:

> The average integrated carbon, nitrogen, and phosphorus signals (n > 1000 for each data point) are shown in Figure 3, along with ordinary least squares linear regressions. (Page 9 Lines 16-17)

> Also illustrated are ordinary least squares linear regressions and reference 1:1 lines indicating equal signal in subsurface and microlayer samples. (Figure 3 Caption)

Please also add a statement on how and where the data can be accessed. I recommend that you upload the data to a public repository which also provides a DOI. Please have a close look at the ACP's Data Policy (https://www.atmospheric-chemistry-and-physics.net/about/data_policy.html).

We have now uploaded relevant data to the Harvard Dataverse repository. We describe access to these datasets in a newly added Data Accessibility section at the end of our manuscript:

Data used to generate this manuscript's figures are included in the Supplemental Information section and a Harvard Dataverse dataset under the name "A Link between the Ice Nucleation Activity and the Biogeochemistry of Seawater." The DOI of this dataset is: 10.7910/DVN/QEJJMF. Further data inquires can be directed to the corresponding author (Daniel Cziczo, djcziczo@purdue.edu).

**Reviewer 1**

The manuscript has improved substantially. All my concerns were removed.

We again thank the reviewer for their suggestions to improve our manuscript.

There is only one small thing: particle mobility does not scale linearly with particle diameter, and doubly charged 200nm particles have a size of ~ 330nm, triply charged ones of ~ 450nm. Based on your size distribution, this will indeed not be a huge problem. But please correct that in your text, at and around line 6, page 7 (in the version without tracked changes).

We have corrected this error in the revised manuscript, now referring explicitly to double and triply charged particles and their approximate diameters. The text now reads:

> Nonetheless, doubly charged particles may have been sampled. Figure S2 illustrates that the concentrations of doubly and triply charged particles (approximately 330 and 450 nm in diameter, respectively) are much lower than singly charged particles. Concentrations of particles with multiple charges were less than 20% of the concentrations of 200 nm particles we size selected. Given that the ratio of doubly to singly charged particles predicted by a Fuchs charging model applied to a DMA neutralizer is below 0.3 (Mamakos, 2016), we estimate that multiply charged particles constitute only less than 6% of the total particles sampled. (Page 7 Lines 6-11)

**Reviewer 2**

The authors have done a decent job editing the manuscript to remove the major issues I identified during the first round of review. However, given the extended nature of these edits it is my view that the title of the manuscript should be changed to more accurately reflect the content. As such, it is my view that the manuscript should be reconsidered after major revisions. I would suggest a title such as "A link between the ice-nucleating activity of seawater samples and their biogeochemistry" or something similar.

We thank the reviewer for recommending we alter the title to better reflect our study's findings. We have adapted the title to remove reference to sea spray aerosol. We believe our new title, "A Link between the Ice Nucleation Activity and the Biogeochemistry of Seawater," more accurately summarizes the scope of the manuscript.

One other comment I have is that it is unnecessary to waste a figure in the main text to present the aerosol size distribution produced by the atomiser. When I asked for this I had assumed the authors should put it into the SI where I think it is better placed (it is critical supplementary information and not a main finding from the work).

We apologize for misunderstanding the original request of the reviewer. We agree that the figure depicting the size distribution of the atomizer is best suited for placement in the Supplemental Information section. We have moved it there and renumbered the figures accordingly.

[revised manuscript text omitted]